

# The BErkeley Atmospheric CO₂ Observation Network: initial evaluation

Alexis A. Shusterman[1], Virginia Teige[1], Alexander J. Turner[2], Catherine Newman[1], Jinsol Kim[1], Ronald C. Cohen[1,3]

[1]Department of Chemistry, University of California Berkeley, Berkeley, CA 94720, USA
[2]School of Engineering and Applied Sciences, Harvard University, Cambridge, MA 01451, USA
[3]Department of Earth and Planetary Science, University of California Berkeley, Berkeley, CA 94720, USA

*Correspondence to*: Ronald C. Cohen (rccohen@berkeley.edu)

**Abstract.** With the majority of the world population residing in urban areas, attempts to monitor and mitigate greenhouse gas emissions must necessarily center on cities. However, existing carbon dioxide observation networks are ill-equipped to resolve the specific intra-city emission phenomena targeted by regulation. Here we describe the design and implementation of the BErkeley Atmospheric CO₂ Observation Network (BEACO₂N), a distributed CO₂ monitoring instrument that utilizes low-cost technology to achieve unprecedented spatial density throughout and around the city of Oakland, California. We characterize the network in terms of four performance parameters−cost, reliability, precision, and bias−and find the BEACO₂N approach to be sufficiently cost-effective and reliable while nonetheless providing high-quality atmospheric observations. First results from the initial installation successfully capture hourly, daily, and seasonal CO₂ signals relevant to urban environments on spatial scales that cannot be accurately represented by atmospheric transport models alone, demonstrating the utility of high-resolution surface networks in urban greenhouse gas monitoring efforts.

## 1 Introduction

As two-thirds of the human population stand to inhabit cities by 2050 (United Nations, 2014), developing a thorough understanding of urban greenhouse gas emissions is of ever-growing importance. International and local law-making bodies around the world are agreeing to caps on total emissions and enacting multi-faceted regulations aimed at achieving these caps (e.g., A.B. 32, 2006; United Nations, 2015). As of yet there exists no mechanism for judging the efficacy of these individual rules or verifying compliance through direct observations of changes in CO₂ at the scale of cities (Duren et al., 2012).

Traditional strategies for assessing greenhouse gas emissions are limited to a small handful of monitoring instruments scattered sparsely across remote areas, mostly in developed nations (e.g., Worthy et al., 2003; Thompson et al., 2009; Andrews et al., 2014). These stations are capable of measuring regional average concentrations with extreme accuracy and precision, but are purposefully distanced from and experience reduced sensitivity to urban signals, thus giving no information on emissions in the precise areas that the majority of greenhouse gas rules aim to regulate.





The increasing significance of urban emissions combined with the proliferation of commercial cavity ring-down spectroscopic instrumentation have resulted in a recent trend towards network sensing approaches for constraining greenhouse gas emissions in cities. For example, Ehleringer et al. (2008) maintain a $CO_2$ monitoring network in the Salt Lake City metropolitan area, the INFLUX network measures $CO_2$, $^{14}CO_2$, and total column $CO_2$ across the city of

5 Indianapolis (Turnbull et al., 2015), and NASA's Megacities Carbon Project has established sensor networks in the pilot cities of Los Angeles (Kort et al., 2013) and Paris (Bréon et al., 2015). These ground-based monitoring efforts are complemented by space-based observations from SCIAMACHY (Burrows et al., 1995), GOSAT (Yokota et al., 2009), and most recently the Orbiting Carbon Observatory-2 (OCO-2) launched in July 2014, which provides total column $CO_2$ measurements over 1.29 by 2.25 km footprints once every 16 days (Eldering et al., 2012).

Thus far, the urban surface projects have relied on a relatively small number of instruments (between 5 and 15) distributed with sensor-to-sensor distances of 5 to 35 km. Initial efforts suggest this approach may be effective at characterizing average citywide emissions over monthly to annual timescales (McKain et al., 2012), however it has yet to be used to identify and quantify specific emission activities at neighborhood scales. To resolve individual emission sources, much finer spatial resolution is needed. Simple Gaussian dispersion models with total reflection at the surface predict >95%

of the one-dimensional footprint of a sensor 10 m above ground level to be within 1.1 km of the sensor under typical conditions (Seinfeld and Pandis, 2006) and prior studies (e.g., Zhu et al., 2006; Beckerman et al., 2007; Choi et al., 2014) have observed e-folding distances of ~100 to 1000 m for urban pollutant plumes mixing into the local background.

Here we propose an alternative approach that strikes a different balance between instrument quality and quantity than in previous $CO_2$ monitoring efforts. The BErkeley Atmospheric $CO_2$ Observation Network (BEACO$_2$N) is a large-scale

network instrument that aims to leverage low-cost sensing techniques in order to enable a spatially dense network of $CO_2$ sensing "nodes" in and around the city of Oakland, California (Fig. 1 and 2). Using commercial $CO_2$ instrumentation of moderate quality and a suite of low-cost trace gas sensors for additional source attribution specificity, BEACO$_2$N is able to achieve an unprecedented spatial resolution of approximately 2 km–to our knowledge the only sensor network to date that monitors $CO_2$ on scale with the heterogeneous patterns of urban sources and sinks (see Fig. 3 for examples of observed intra-

city $CO_2$ gradients). We present an initial description and characterization of the instrument, beginning with a description of the nodes, their locations, and the development of various laboratory and in situ calibration techniques. We then evaluate the network in terms for four factors–cost, reliability, precision, and bias; described below–and conclude by demonstrating BEACO$_2$N's ability to resolve $CO_2$ signals of significance to the urban environment.

## 1.1 Cost

In order to remain cost-competitive with other, less dense networks employing higher-grade instrumentation, a high-density network must utilize sensors with a price 1–2 orders of magnitude lower. However, as sensor price often scales with quality, low-cost instrumentation may carry associated penalties in other domains, such as diminished precision, persistent bias, or



the need for frequent maintenance and/or re-calibration. Thus, we seek to optimize the trade-off between cost and the other considerations.

## 1.2 Reliability

Network reliability consists of sensor uptime and continuity of the data stream and is crucial to enabling comparison and averaging across sites as well as improving the statistics of temporal analyses. Poor reliability also has an indirect impact on cost via the resources expended on repeat maintenance visits and/or replacement part purchases.

## 1.3 Precision

The precision requirements at each individual site versus for a network instrument as a whole vary depending on the phenomena of interest. For the characterization of inter-annual trends integrated over entire metropolitan regions, sensitivity to changes <10 ppm per year are required (Pacala et al., 2010). For example, according to the First Update to the Climate Change Scoping Plan, the state of California would have to reduce its overall $CO_2$ emissions by 4.7 million metric tons per year to achieve its goal of reaching 1990 emission levels by 2020 (Brown et al., 2014). Assuming a fraction of that total reduction is attributable to the San Francisco Bay Area in proportion to its population (~20% of the California total), this amounts to a change of $-2.6 \times 10^6$ kg $CO_2$ day$^{-1}$ for the San Francisco Bay Area. Given a residence time of air in the region of 1 day, these emissions reductions spread evenly over the 22,681 km$^2$ domain and through a 1 km boundary layer would lead to a 65 ppb annual decrease in the daily $CO_2$ concentrations. We therefore require $N$ instruments of sufficient individual sensitivity such that, upon averaging their signals together, the resultant $\sqrt{N}$ improvement in the network-wide mean sensitivity allows us to detect annual changes of ~65 ppb year$^{-1}$ with confidence.

However, the true strength of the high-density approach lies in the individual sensors' (or sub-group of sensors') sensitivity to intra-city phenomena, which are orders of magnitude larger by virtue of their proximity to sources not yet diluted by advection. Larger signal sizes forgive poorer precision in the individual instruments, but demand sufficient temporal resolution to capture these anomalous, often unexpected, events of short duration on top of slowly varying domain-wide fluctuations in the background concentration. Because the BEACO$_2$N instrument is unique in its sensitivity to these highly local processes, we will focus on this latter specification of the instrument precision in the characterization that follows.

## 1.4 Bias

Bias can be incurred somewhat abruptly during the initial field installation or accrued more gradually as drift away from the laboratory calibration over time. Systematic bias in the sensor readings is of particular concern in a large-scale network deployment where onsite calibration materials such as reference gases are infeasible and frequent maintenance visits are undesirable. To ensure trustworthy observations, a given network sensing approach must demonstrate some combination of:

(a) instrumentation that is sufficiently robust against sudden or gradual introduction of bias in excess of the instrument precision, (b) a post hoc correction for bias in the data record, and/or (c) a procedure for identifying and replacing sensors whose bias cannot be remedied via the prior methods.

## 2 Node Design, Calibration, and Deployment

Each BEACO2N node contains a non-dispersive infrared Vaisala CarboCap GMP343 sensor for $CO_2$ as well as SGX Sensortech MiCS-4514 and MiCS-2614 metal oxide-based micro-sensors used to detect $CO/NO_2$ and $O_3$, respectively. Following a large-scale node refurbishment and upgrading effort in mid-2014, these core elements are now supplemented with a Sensirion SHT15 and Bosch-Sensortec BMP180 for measuring humidity (SHT15), pressure (BMP180), and temperature (both), a Shinyei PPD42NS nephelometric particulate matter sensor, and a suite of Alphasense B4

electrochemical trace gas sensors for $O_3$, CO, NO, and $NO_2$. Discussion of these latter, air quality-related technologies will follow in a forthcoming paper.

    All sensors are assembled into compact, weatherproof enclosures as seen in Fig. 4. A Raspberry Pi microprocessor automates data collection via a serial-to-USB converter (for $CO_2$, every ~2 seconds) and an Arduino Leonardo microcontroller (for everything else, every ~10 seconds), then transmits data to a central server using either: (a) a direct on-

site Ethernet connection, (b) a Ubiquiti NanoStation locoM2 Wi-Fi antenna, or (c) an Adafruit FONA miniGSM cellular module. The latter has the unintended consequence of introducing a significant amount of electrical noise into the system. We reduce the impact of this noise by limiting data transmission to two hours per day, on a rotating schedule such that no periods are disproportionately afflicted by elevated noise levels. Battery-powered real time clock modules are also included to ensure timestamp accuracy during planned and unexpected hiatuses in internet connectivity.

Airflow through the node is maintained by two 30 mm fans, one positioned in the "intake" orientation and the other in the "outflow" orientation. An additional, passive air outlet is located adjacent to the AC/DC power supply converter to prevent excessive heating inside the node. Node enclosures measure 90 mm by 160 mm by 360 mm and are made of corrosion-resistant die cast aluminium that minimizes meteorological and magnetic complications. Stainless steel fasteners and a weatherproof seal prevent water intrusion into the enclosure.

Laboratory calibrations are performed on each CarboCap sensor upon receipt of the instrument from the supplier and repeated whenever nodes are retrieved from the field for maintenance, resulting in a re-calibration every 12–18 months. Reference cylinders of 0 ppm, 1000 ppm, and either 320 ppm or 370 ppm ($\pm 1\%$) are used for ~10 minute deliveries of each concentration to a chamber containing the sensor, which includes a built-in microprocessor that accepts the results of this multi-point calibration as input and automatically applies the appropriate corrections to the subsequent observations. The

CarboCap microprocessor can also be configured to correct for the effects of oxygen, temperature, pressure, and humidity. The built-in oxygen compensation is utilized at a constant value of 20.95%, while the latter three compensations are turned



off prior to sensor deployment. Instead, a post hoc correction is derived from the ideal gas law and Dalton's law of partial pressures:

$$[CO_2]_{dry} = [CO_2]_{raw} * \frac{1013.25\ hPa}{P_{tot}} * \frac{T}{298.15\ K} * \frac{1}{1-(\frac{P_{H_2O}}{P_{tot}})} \tag{1}$$

Here $[CO_2]_{dry}$ is the dry air mole fraction, or the amount of $CO_2$ that would be measured if the observed air parcel was dried

and brought to standard temperature and pressure. $[CO_2]_{raw}$, $T$, $P_{tot}$, and $P_{H_2O}$ are, respectively, the raw $CO_2$ concentration output by the CarboCap software in ppm, the temperature measured by the internal thermometer of the CarboCap in K, the atmospheric pressure in hPa, and the partial pressure of water in hPa, derived from the dew point temperature ($T_{dew}$, in ºC) using the August-Roche-Magnus approximation of the Clausius-Clapeyron relation as indicated below:

$$P_{H_2O} = 6.1094\ hPa * exp(\frac{17.625\ T_{dew}}{243.04+T_{dew}}) \tag{2}$$

For post-2014 observations, we use the pressure and dew point temperature measured inside each node enclosure by the aforementioned BMP180 and SHT15 sensors, respectively. For data collected prior to 2014, Eq. (1) and (2) are calculated from the average sea level pressures (adjusted for altitude) and dew point temperatures measured within ~50 km of the BEACO2N domain by weather stations in the NOAA Integrated Surface Database (https://www.ncdc.noaa.gov/isd/).

Figure 5 compares 1 minute mean $CO_2$ dry air mole fractions calculated as described above with readings from a custom

cavity ring-down reference instrument based on the Picarro G2301 analyzer system co-located with an in-field CarboCap over the course of 2 weeks in January 2016. The ratios between the CarboCap and Picarro observations are then shown in Fig. 6 as a function of temperature, total pressure, and the partial pressure of water. Although most of the impact of these environmental variables is removed by the ideal gas law-based correction in Eq. (1), slight dependencies on each variable remain, likely due to their influence on the vibrational spectra of $CO_2$ via pressure broadening, etc. Performing similar

analyses on observations from in situ co-locations with other reference instruments (see LI-COR LI-820 in Sect. 3.4) reveals the temperature and water dependence to vary in sign and magnitude between individual sensors, while the pressure dependence is found to be quite robust. We therefore apply the following empirical correction to all $CO_2$ observations with coincident, on-site pressure measurements (i.e. post-2014 data sets):

$$[CO_2]_{corrected} = [CO_2]_{dry} * (-0.00055\ P_{tot} + 1.5) \tag{3}$$

The effect of this correction is shown in the histogram of CarboCap–Picarro differences in Fig. 5 (gray bars). The offset between the two instruments is reduced to from -1 ppm to ~0 ppm and the standard deviation of their differences is tightened from ±1.5 ppm to ±1.4 ppm. The lingering temperature and water biases are <1%.

Calibrated nodes are installed 2 to 111 m above ground level (6 to 476 m above sea level), mounted to existing infrastructure or weighted industrial tripods. Rooftop position and intake orientation are chosen to optimize wireless

connectivity (if applicable), maximize air exchange with the surrounding area, and minimize sampling of extremely local emission sources (e.g., rooftop ventilation ducts). BEACO2N nodes are sited on an approximately 2 km square grid across the Oakland metropolitan area (see Fig. 1 and 2 and Table 1), often on top of schools and museums, which possess roughly



the desired spatial density and also assist the service of the educational and outreach goals of the project (see http://beacon.berkeley.edu). The 2 km spacing is chosen to ensure an approximately 1 km proximity to any significant $CO_2$ source or sink in the metropolitan area, maximizing coverage without undue overlap between neighboring footprints. Additional sites outside the 2 km grid are also included for sensitivity to potential emission sources of interest, for co-

location with useful reference instruments, or as pilots for network expansion.

## 3 Node Performance

### 3.1 Cost

The Vaisala CarboCap GMP343 $CO_2$ sensor in this study is used in its 0 to 1000 ppm measurement range and "diffusion sampling" mode, such that representative air samples passively diffuse into the path of the infrared light beam. With these

specifications, each CarboCap costs approximately \$2,800 USD. Although less expensive technologies are available, the CarboCap design has a clear advantage in that the unit contains a digitally-controlled Fabry-Perot interferometer to switch on (4.26 µm) and off (3.9 µm) of the asymmetric stretching mode of $CO_2$, generating a baseline intensity measurement for each observation that compensates for variability in the light source.

Additional sensors, ancillary hardware, and labor then bring the total cost per node to ~\$5,500 USD, or \$154,000 USD

for the entire 28 node BEACO2N instrument. For comparison, a single commercial cavity ring-down analyzer is priced around \$60,000 USD and the annual operating cost can exceed \$85,000 USD after accounting for pumps, data loggers, etc.

### 3.2 Reliability

Table 2 gives the percent uptime for nine representative BEACO2N nodes over the course of 2013, calculated as the fraction of total minutes in the year during which a given node collected valid data. All nine nodes exhibit uptimes in excess of 50%

via either hardwired Ethernet connections or Wi-Fi antennas, with six collecting data >80% of the time. Maintenance visits to these sites beginning in mid-2014 revealed little to no incidence of hardware failure. Instead, external issues, such as interruptions in the electricity or Wi-Fi connectivity, are found to be the limiting factors in determining sensor uptime. Transplanting nodes to sites with more dependable electricity supplies and increasing implementation of cellular modules (which are insensitive to interruptions in onsite Wi-Fi networks) continue to enhance network reliability over time. For

example, the nine most reliable nodes during the January 2015–April 2016 period all exhibit uptimes >80%, with five collecting data and transmitting it within the next 48 hours ~100% of the time via either Ethernet or cellular data communication.

### 3.3 Precision

From a qualitative perspective, the Vaisala CarboCap GMP343 demonstrates exceptional sensitivity to $CO_2$ enhancements

on scales typical of an urban environment. Figure 7 compares the 1 minute mean $CO_2$ dry air mole fractions measured at two





nearby in-field BEACO2N nodes (EXB and EXE in Fig. 1) during 1 week in early October 2015. As these sensors are not precisely co-located (one is stationed approximately 13 m above roadside in downtown San Francisco, while the other sits ~200 m back from the road, near the bay), an exact correlation is not expected. The two sensors nonetheless demonstrate remarkable agreement; while $CO_2$ variations resulting from boundary layer height fluctuations and wind speed changes during the same period are on the order of 20–60 ppm, the CarboCaps simultaneously detect $CO_2$ events as small as 8 ppm, providing preliminary evidence of the suitability of these sensors for high-density urban deployment.

More quantitatively, Vaisala advertises the CarboCap as possessing a response time of 75 seconds and a precision of ±3 ppm at 2 second measurement frequency. Here we present our own characterization of the sensors' precision via comparison to: (a) in-laboratory reference gases and (b) a co-located in situ reference instrument.

After exposing an ensemble of CarboCaps to a constant stream of reference gas, we find the 1 minute mean dry air mole fractions to exhibit $1\sigma$ precision between ±1.2 and ±2.0 ppm, roughly in keeping with the ±2 ppm precision observed by Rigby et al. (2008). Figure 5 shows the results from our aforementioned co-location with a Picarro G2301 reference instrument, demonstrating near perfect correlation ($R^2$ = 0.9999), slope $\cong$ 1, and an offset of approximately 0 ppm after meteorological corrections. In this case the $1\sigma$ precision is ±1.4 ppm, given directly by the standard deviation of the differences between the minute-averaged CarboCap and Picarro observations, as the Picarro's precision (±0.1 ppm) contributes negligibly to the noise. This presents a slight improvement over the ±2.18 ppm in situ precision recorded by van Leeuwen et al. (2010), although still greater variability than would be expected given the manufacturer's 2 second specifications and a 1 minute averaging time ($3\ ppm/\sqrt{30} = 0.55\ ppm$). Nonetheless, the agreement between the time series of the Picarro and CarboCap measurements demonstrates this noise level to be effectively negligible on the scale of ambient urban $CO_2$ fluctuations. Furthermore, we see that averaging the observations from 28 CarboCaps with ±1.4 ppm precision easily produces the ~65 ppb sensitivity required to detect subtler network-wide fluctuations occurring across multiple years (see Sect. 1.3).

## 3.4 Bias

Given the limited access to validation and calibration infrastructure, a major concern for a long-term field deployment is bias resulting from a combination of gradual temporal drift ($B_{temporal}$, in ppm day$^{-1}$) and constant offsets from the "true" value ($B_{atemporal}$, in ppm), perhaps incurred abruptly upon installation. The measurement at a given site ($[CO_2]_{node}$, in ppm) is therefore the sum of the real regional and local influences at said site ($[CO_2]_{background}$ and $[CO_2]_{local}$, respectively), as well as these spurious biases:

$$[CO_2]_{node} = [CO_2]_{background} + [CO_2]_{local} + B_{atemporal} + \left(B_{temporal} \times days\right) \tag{4}$$

To derive post hoc corrections for $B_{atemporal}$ and $B_{temporal}$ at a given site, we first remove the $[CO_2]_{background}$ signal from the data record by subtracting off the weekly minimum $CO_2$ concentrations recorded at a reference site within the network domain. BEACO2N's unique location near the Pacific coast results in a relatively consistent wind direction from



largely unpolluted over-ocean origins, such that the weekly minima reflect both the seasonal and synoptic variations in network-wide baseline $CO_2$ concentrations while avoiding the influence of shorter term variability in local sources and sinks. Once the $[CO_2]_{background}$ term is removed, we calculate the weekly minima of the de-seasonalized data record and fit the result as a (piecewise, if necessary) linear function of time, the slope of which gives the value of $B_{temporal}$. This linear fit is

5 then itself subtracted from the de-seasonalized data record, yielding a remainder comprised of only the $[CO_2]_{local}$ and $B_{atemporal}$ terms. While the $[CO_2]_{local}$ component varies rapidly over several orders of magnitude, the contribution of $B_{atemporal}$ is, by definition, constant in time, so we once again compute the weekly minima of the new data record and define the mean weekly minimum as $B_{atemporal}$. Having obtained values for $B_{temporal}$ and $B_{atemporal}$, we simply subtract these components from the original data record to generate the unbiased observations at each site.

10 Table 3 gives the results from one iteration of the bias correction outlined above executed using the ELC BEACO2N node (see Fig. 1) as the reference site needed to calculate $[CO_2]_{background}$. Only sites that enable at least 3 months of comparison to the ELC node are included; multiple values at a single site correspond to the piecewise linear fits employed when $B_{temporal}$ exhibits discontinuities over the data record. Because we universally define Day 1 to be 1 January 2013 and $B_{atemporal}$ is strongly influenced by the intercept of the linear fit used to characterize the temporal drift, it should be noted

15 that the magnitude of $B_{atemporal}$ does not necessarily represent the actual bias present at a node on its deployment date (which may be before or after 1 January 2013), but rather an extrapolation of this initial bias forwards or backwards in time.

To evaluate the efficacy of this procedure, we compare the weekly minima of both the raw and bias corrected data records to the weekly minimum $CO_2$ concentrations measured by a LI-COR LI-820 non-dispersive infrared $CO_2$ gas analyzer positioned at sea level between the EXB and EXE nodes (see Fig. 1). The LI-COR is maintained by NOAA's Pacific Marine

20 Environmental Laboratory and calibrated against compressed gas (400–500 ppm $CO_2$) prior to every hourly measurement and is assumed to have negligible bias. The results of said comparison are shown in Fig. 8, demonstrating significantly improved agreement (3.7 vs. 9.8 ppm mean residuals) with the LI-COR weekly minima after bias correction. This is likely a conservative estimate of the bias reduction achievable with this method, as the ELC node we use to compute our $[CO_2]_{background}$ value is not itself a bias-free reference. Although the raw ELC data record demonstrates the least bias of all

25 the BEACO2N nodes in an initial comparison with the LI-COR, its observations are nonetheless afflicted by some unknown nonzero drift and/or atemporal offset. Direct in situ calibration of the reference instrument would allow us to constrain systematic biases further, possibly to within the precision of the sensors.

### 3.5 Performance of Ancillary Sensor Technology

According to manufacturer documentation, the Sensirion SHT15 provides relative humidity measurements to 0.05%

30 resolution, with an advertised accuracy of ±2.0%, a repeatability of ±0.1%, an 8 second response time, and a long-term drift of <0.5% per year. Its temperature measurements are provided to 0.01 °C resolution, with an advertised accuracy of ±0.3 °C, a precision of ±0.1 °C, a response time of 5 to 30 seconds, and a long-term drift of <0.04 °C per year. The Bosch-Sensortec





BMP180 provides pressure measurements to 0.01 hPa resolution, with an advertised accuracy of ±0.12 hPa, a precision of ±0.05 hPa, and a long-term drift of ±1.0 hPa per year. Its temperature measurements are provided to 0.1 ℃ resolution, with an advertised accuracy of ±1.0 ℃. An independent verification of these performance specifications is not attempted here. However, the temperature observations from both sensors closely follow the structure of that detected by the internal temperature sensor of the CarboCap, although the CarboCap's readings are consistently slightly elevated, as expected given the heat produced by the instrument itself.

The BMP180 and SHT15 are not intended to reflect local meteorological conditions, but rather to provide a representative picture of conditions inside the node These internal conditions are integral to various posterior corrections applied to the observations (see Sect. 2).

## 4 Initial Field Results

The BEACO₂N field campaign is a long-term, ongoing monitoring effort. Here we provide a time series of data collected from 17 BEACO₂N sites between January 2013 and April 2016 (Fig. 9) and some initial descriptive statistics of the bias-corrected dry air $CO_2$ mole fractions at nine representative sites in 2013 (Table 2).

Figure 9 demonstrates the extreme short-term variability in urban $CO_2$ concentrations superimposed on a slower, seasonal fluctuation in the minimum values. Daytime (1100–1800 LT) means between 408 and 442 ppm are observed, with maximum values between 500 and 820 ppm and minima between 384 and 396 ppm. Standard deviations are seen to range from 9.57 to 22.4 ppm, all of which are lower than the corresponding nighttime (2200–0400 LT) standard deviations due to the reduced convective mixing in the shallow nocturnal boundary layer. Similarly, the majority of nighttime means and maxima exceed the daytime values at the same location, with the exception of three sites: FTK, LAU, and KAI. The dampened or inverted diurnal trends at these sites may be due to unique boundary layer dynamics at those locales or unusually large daytime $CO_2$ sources nearby. Daytime and nighttime minima do not differ as significantly.

Individual BEACO₂N nodes are observed to capture a number of patterns and cycles typical of ambient $CO_2$ monitoring. Figure 10 shows the monthly minimum $CO_2$ concentrations at six select sites in 2013, as the difference from their July value (defined as 0 ppm at each site). A distinct seasonal cycle is observed, with wintertime minima exceeding summertime values by 7 to 24 ppm. For reference, the gray curve presents a similar treatment of a smoothed, three-dimensional "curtain" of surface $CO_2$ Pacific boundary conditions produced by NOAA's Global Greenhouse Gas Reference Network (Jeong et al., 2013). In the summertime, the BEACO₂N minima are seen to converge to a seasonal variation roughly in keeping with that observed in the curtain, while the degree of variability within the network increases during the winter months.

Figure 11 shows representative diurnal cycles in the bias-corrected $CO_2$ dry air mole fractions at three different BEACO₂N nodes in September 2013. We observe elevated concentrations at night corresponding to a shallow nocturnal boundary layer, significant enhancements around the morning rush hour when emissions are increasing faster than boundary layer height, and midday minima reflecting mixing into the largest volume of air before the boundary layer collapses again at





sunset. However, within this qualitatively well understood average behavior remains a degree of intra-network variability that allows us to discover and probe local scale phenomena of unknown origin. At FTK, for example, concentrations are seen to decrease after an initial rush hour peak (~0800 LT) but remain somewhat elevated until sunset, never achieving the much more pronounced afternoon minimum observed at PAP, 13.5 km away.

Such intra-city heterogeneities are difficult to capture accurately using atmospheric transport models alone. We calculate the spatial footprint of each site using a Stochastic Time-Inverted Lagrangian Transport Model (Lin et al., 2003) coupled to the Weather Research and Forecasting model (Skamarock et al., 2008) (WRF–STILT) in the manner of Nehrkorn et al. (2010) and overlay it on a high-resolution bottom-up emissions inventory (see Fig. 3) as in Turner et al. (2016) to generate the predicted diurnal cycles shown as black squares in Fig. 11. While the model captures midday conditions at FTK and

evening levels at PAP, the presence of both over- and under-estimations at other times suggests a need to re-examine the bottom-up emissions inventory as well as the model's treatment of boundary layer dynamics. BEACO$_2$N provides the ground truth necessary to identify such deficiencies and potentially improve upon them via inverse modeling, data assimilation, etc.

Comparison of diurnal cycles during noteworthy local scale emission events with averages such as those seen in Fig. 11

gives further insight into the potential application of BEACO$_2$N observations to CO$_2$ source attribution. Figure 12 offers one such comparison using hourly averages collected from a BEACO$_2$N node positioned on top of Oakland High School (OHS in Fig. 1 and Table 1) during a weather-related school closure that occurred on 11 December 2014. Clear reductions in CO$_2$ concentrations are observed relative to what is typical at this site (and indeed network-wide, although to a lesser extent), as is expected in the absence of emissions related to students' commutes and presence on campus. The sensing technology

implemented in the BEACO$_2$N nodes therefore proves adequate to resolve not only CO$_2$ patterns typical of an urban environment, but also short-term deviations during anomalous emission events, positioning BEACO$_2$N as an essential tool for the characterization of current urban conditions as well as the verification of subsequent emissions reductions.

## 5 Discussion & Conclusion

We have described the design, implementation, and initial observations from a novel high-resolution CO$_2$ surface monitoring

network instrument. We demonstrate that low-cost instrumentation enables an unprecedented level of spatial density, and describe a calibration protocol with post hoc bias corrections that permit the network to operate precisely and reliably enough to characterize variations in ambient concentrations with magnitudes relevant to metropolitan life.

Our preliminary analysis of the first ~3 years of CO$_2$ observations provides evidence of the expected diurnal and seasonal cycles as well as an encouraging sensitivity to short-term changes in local emission events. Furthermore, we show

significant qualitative and quantitative differences among the diurnal cycles at individual nodes on spatial scales that cannot yet be accurately captured by atmospheric transport models, confirming the necessity of a high-density approach when attempting to faithfully represent the variability of a complex urban environment.





Future work will focus on constructing inferred emissions patterns and trends from the body of observations. In an initial effort in this regard, Turner et al. (2016) constructed and applied a WRF–STILT inverse model to synthetic observations with density similar to BEACO$_2$N. For an area source the size of the Oakland metropolitan area, emissions were estimated to within 18% accuracy; for a freeway-sized line source to within 36%; and to within 60% for the sum of six

industrial point sources—consistently outperforming a smaller hypothetical network (three sites) with significantly better precision. Using week-long averages, the BEACO$_2$N-like network was able to further reduce the uncertainty in the integrated urban area source to <2%, a significant improvement over the citywide emissions estimates provided by real and proposed ~10 site sensor networks described by Lauvaux et al. (2016) (25% uncertainty in 5 day averages), Kort et al. (2013) (>10% uncertainty in monthly averages), and Wu et al. (2015) (10% uncertainty in annual averages). These other studies use more

conservative estimates of the combined instrument, model, and representativeness error ($\geq 3$ ppm, as opposed to $\pm 1$ ppm). These combined error budgets are typically dominated by transport (model) error, which potentially explains why models based on BEACO$_2$N-like networks perform comparably to or better than those based on sparser networks of higher quality sensors, for which instrument error may be reduced but accurately representing transport between observation sites is of greater importance. Further work is needed to fully assess the efficacy of inverse methods based on the BEACO$_2$N approach.

In addition, further characterization of the trace gas and particulate matter sensors will allow for more specific source attribution via the exploitation of emissions factors unique to various combustion activities (e.g., Ban-Weiss et al., 2008; Harley et al., 2015), while providing public health-relevant air quality information as well. There is also potential for fine-grained verification of space-based observations or even of personal sensors when their inherent mobility brings them within the geographical area well represented by the fixed BEACO$_2$N network.

This work constitutes a promising initial infrastructure upon which further advances in high-density atmospheric monitoring can be built, capable of providing research, regulatory, and layperson communities with greenhouse gas and air toxics information on the scale at which emissions and personal exposure actually occur. We are currently planning to expand this validated pilot network into the neighboring locales of San Francisco and Richmond, California, allowing us to characterize other emissions sources, such as oil refining facilities. These efforts will be complemented by modeling studies

comparing different sampling resolutions (i.e. 2 km vs. 4 km sensor spacing) and spatial configurations, yielding general network optimization principles that will facilitate future implementations of high-resolution CO$_2$ monitoring networks in diverse locations.

*Acknowledgements.* This work was funded by the National Science Foundation (1035050; 1038191), the National

Aeronautics and Aerospace Administration (NAS2-03144), and the Bay Area Air Quality Management District (2013.145). Additional support was provided by a NSF Graduate Research Fellowship to AAS, a Department of Energy Computational Science Graduate Fellowship to AJT, a Kwanjeong Lee Chonghwan Educational Fellowship to JK, and the UC Berkeley Miller Institute to RCC. We acknowledge the use of datasets maintained by NOAA's Integrated Surface Database, Pacific Marine Environmental Laboratory, and Global Greenhouse Gas Reference Network. We thank UC Berkeley Research





Computing for access to computation resources, Sebastien C. Biraud for facilitating inter-comparisons with the Picarro G3201, as well as David M. Holstius and Holly L. Maness for their generous contributions to BEACO$_2$N's code base.

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





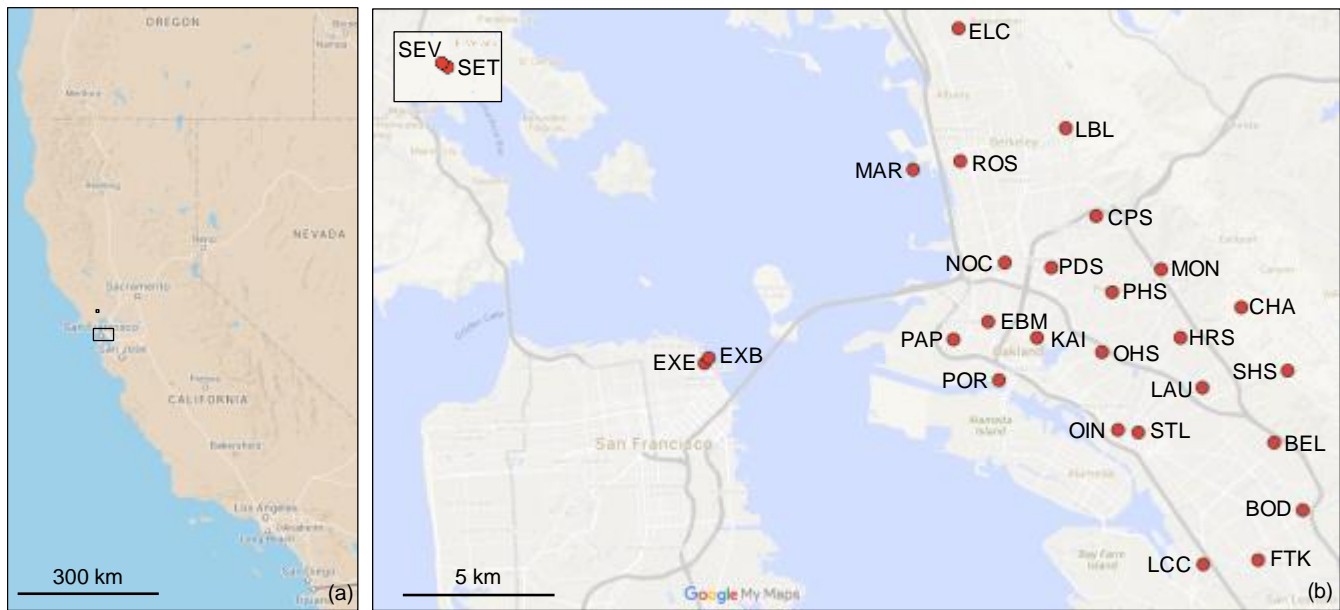

**Figure 1: Map of the BEACO2N domain (a) in the context of the western United States and (b) showing individual node locations. Inset in panel (b) shows the pair of nodes stationed in Sonoma County.**

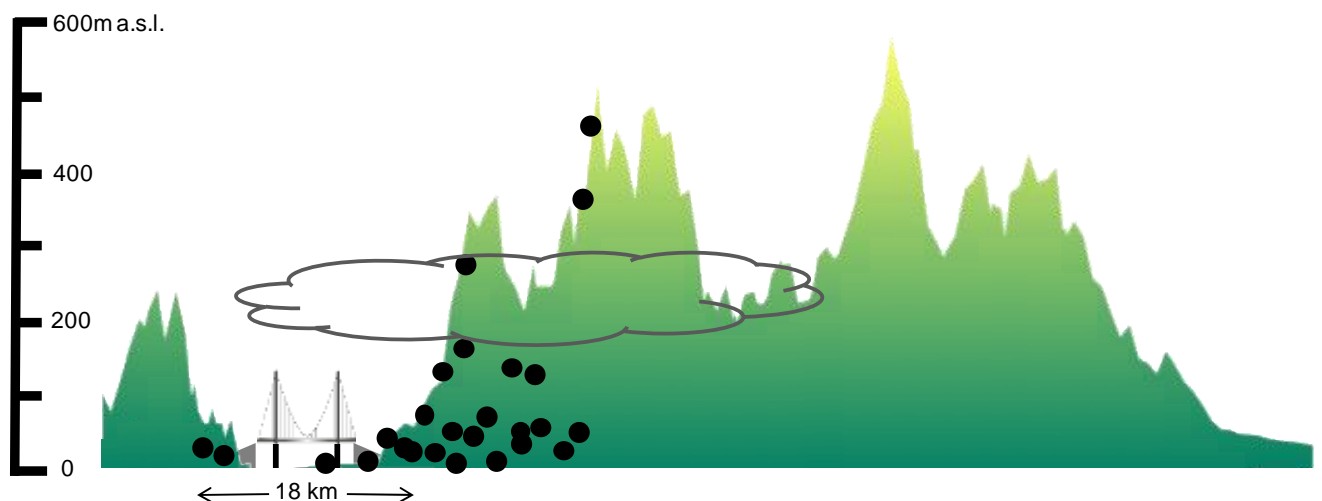

**Figure 2: North-facing schematic of Fig. 1 indicating a representative vertical distribution of BEACO2N node sites (circles) over the topography of Oakland, CA. Cloud marks the altitude and thickness of a typical marine fog layer; bridge delineates the height**

10 **of the San Francisco-Oakland Bay Bridge. Horizontal placement of nodes has been skewed for visual clarity.**



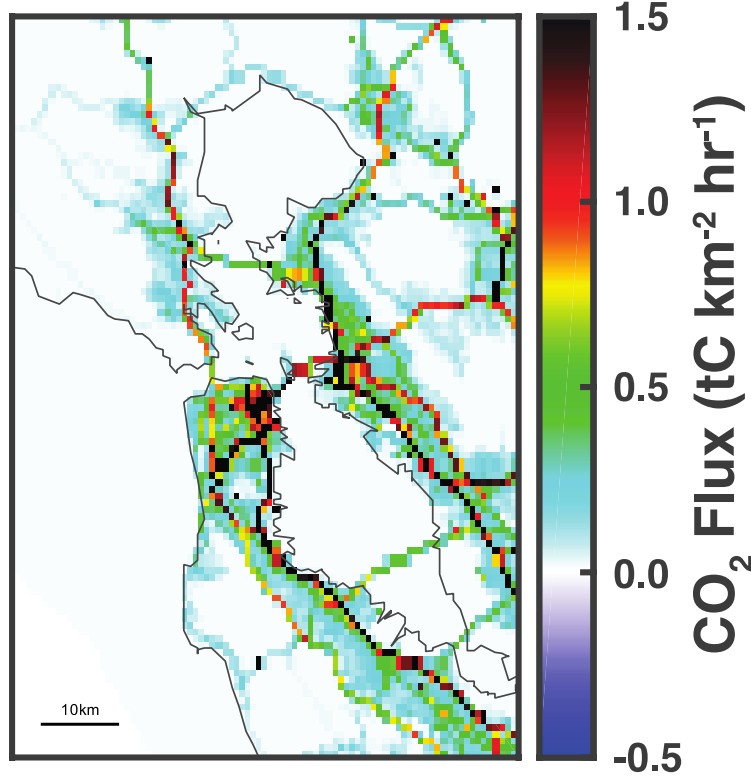

**Figure 3: A sample high-resolution bottom-up emissions inventory for the Bay Area adapted from Turner et al. (2016).**

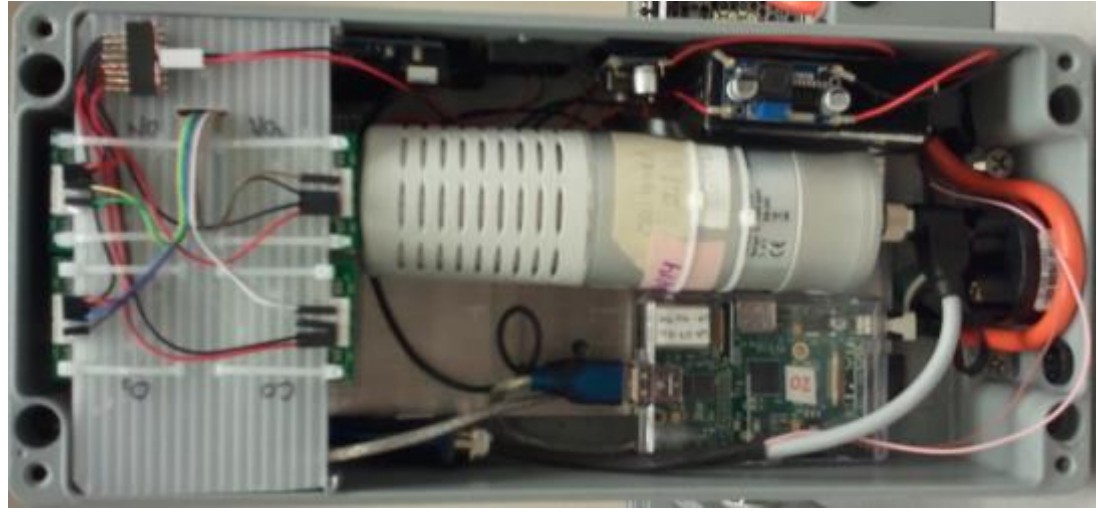

**Figure 4: Current BEACO₂N node design.**





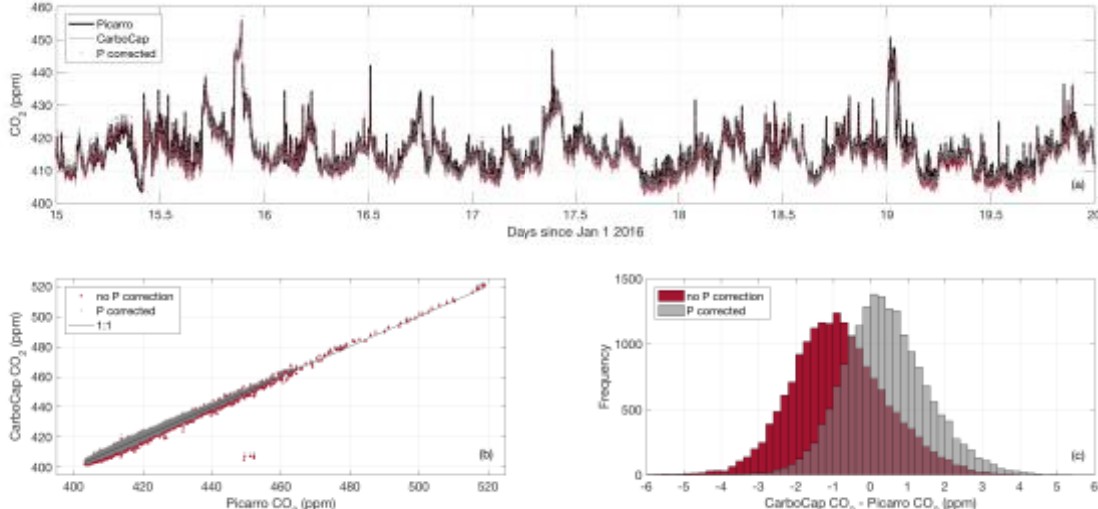

**Figure 5: One minute mean results from a 2 week co-location of a Vaisala CarboCap GMP343 and a custom cavity ring-down reference instrument based on the Picarro G2301 system: (a) representative 5 day time series, (b) direct comparison, (c) histogram of the differences. CarboCap observations are dry air mole fractions calculated using Eq. (1) and subsequently pressure corrected with Eq. (3).**

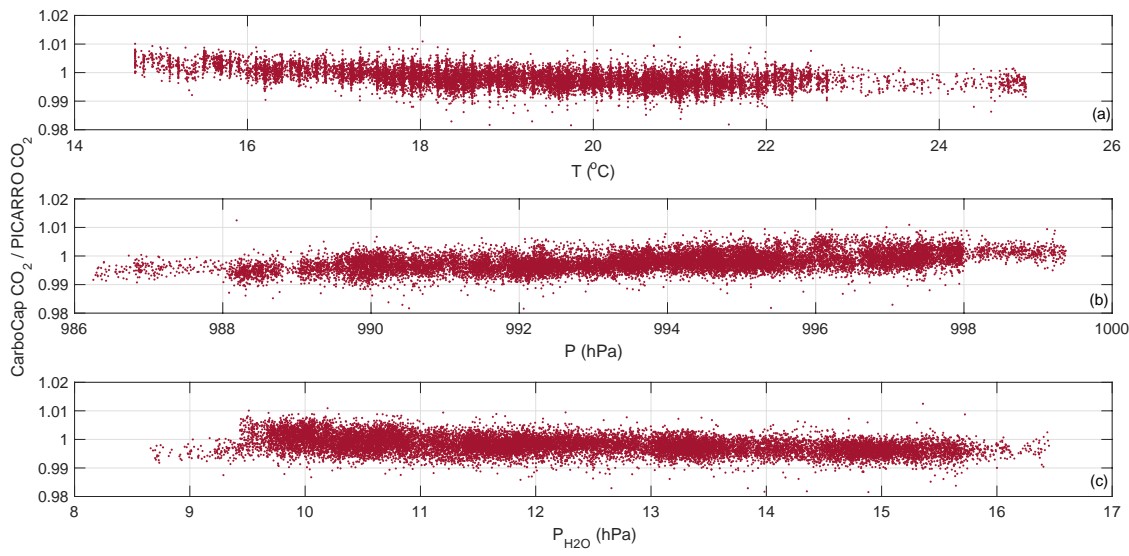

**Figure 6: Ratio of 1 minute mean $CO_2$ dry air mole fractions presented in Fig. 5, shown as a function of temperature (a), pressure (b), and the partial pressure of water (c).**





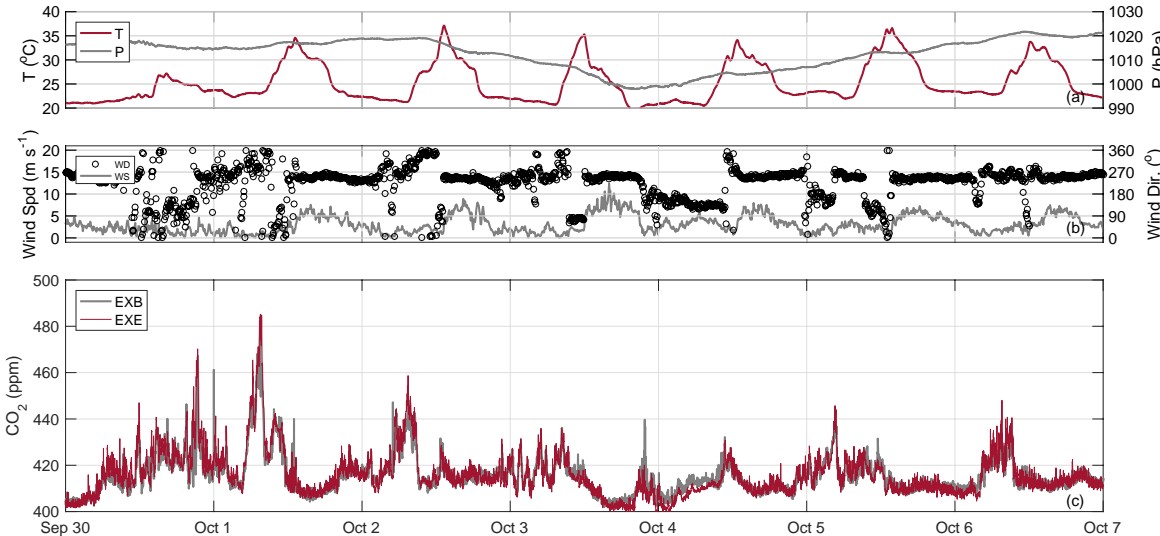

**Figure 7: Representative weeklong time series of observations collected at or near two nearby in-field BEACO₂N nodes (EXB and EXE in Fig. 1; ~200m apart) in October 2015: (a) temperature and pressure averaged to 1 minute, (b) wind speed and direction collected once every 6 minutes, (c) bias-corrected CO₂ dry air mole fractions averaged to 1 minute.**

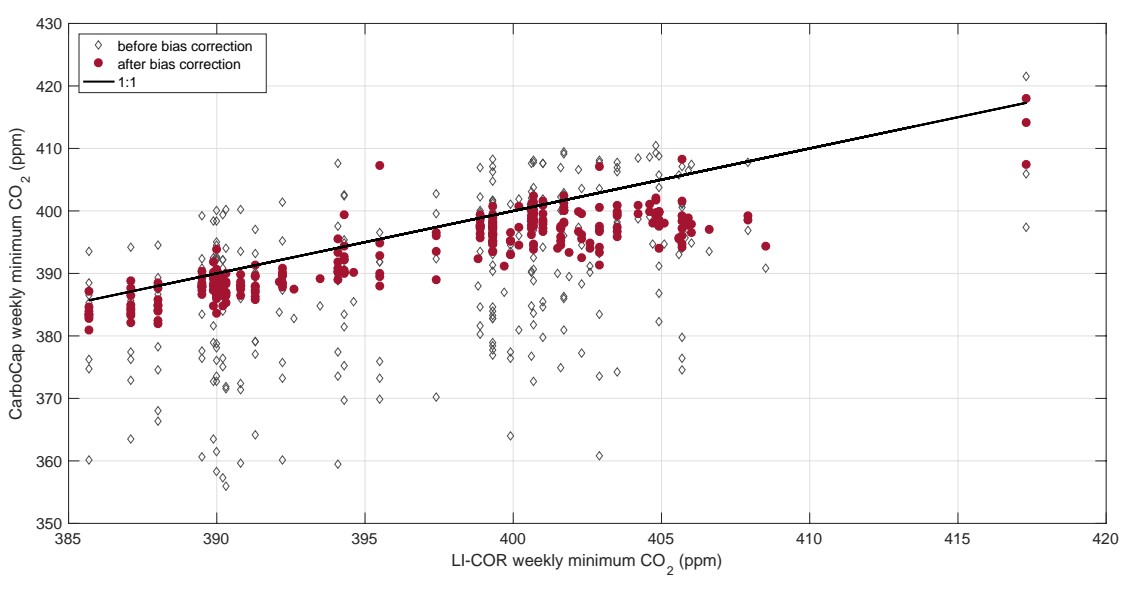

**Figure 8: Weekly minimum CO₂ concentrations measured by a LI-COR LI-820 reference instrument compared with weekly**
10 **minima calculated from the BEACO₂N data record before and after bias correction.**





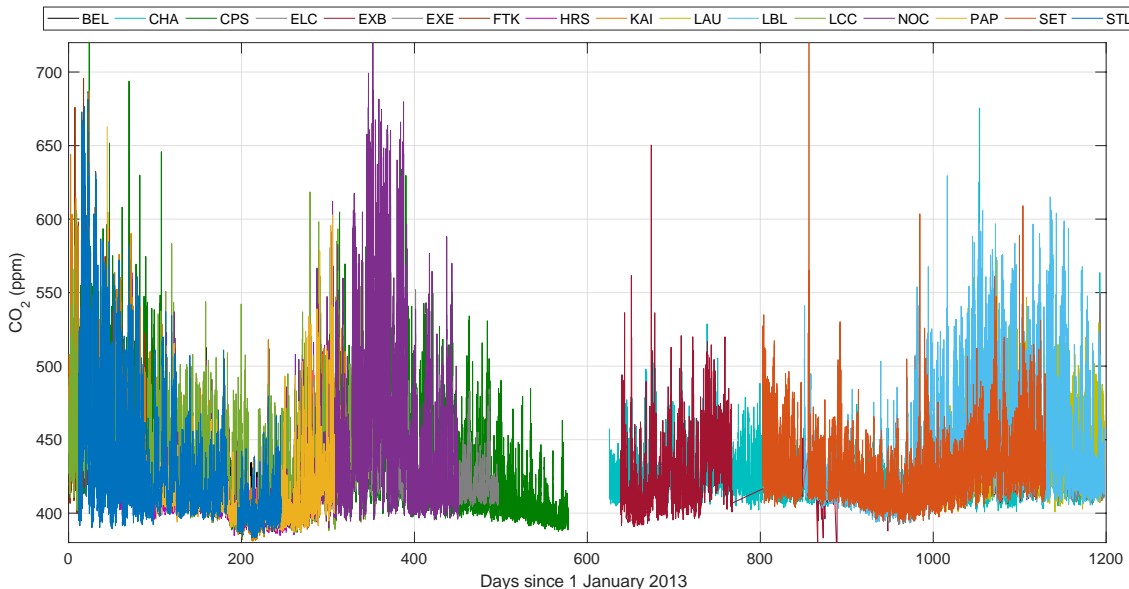

**Figure 9: Time series of bias-corrected $CO_2$ dry air mole fractions from 17 BEACO2N sites. The hiatus around Day 600 corresponds to a largescale hardware refurbishment effort that began in mid-2014.**

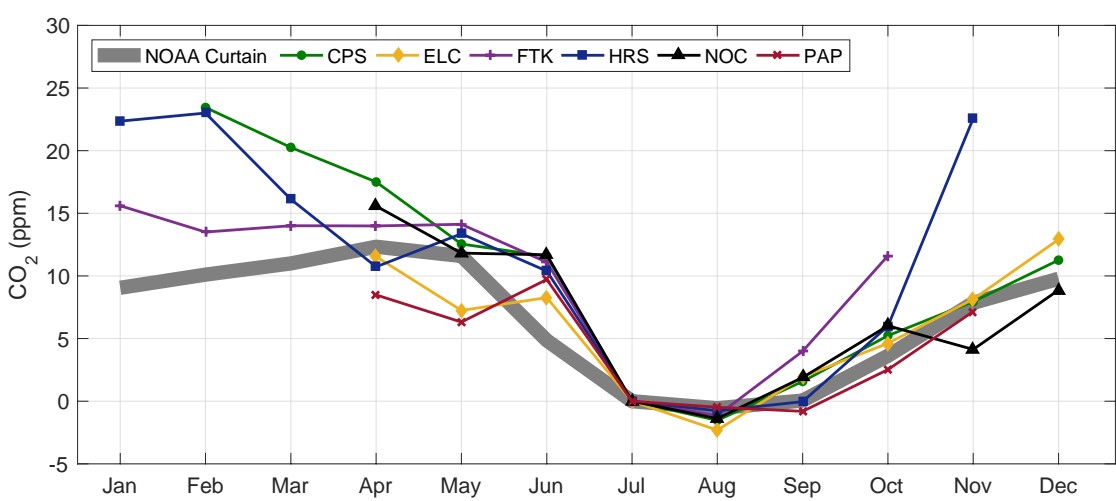

**Figure 10: Monthly minimum bias-corrected $CO_2$ dry air mole fractions observed during 2013 at six representative BEACO2N**
10 **sites, plotted as the enhancement above the July value. Bold gray curve shows a similar treatment of the surface level Pacific Ocean empirical boundary curtain values for 38° N.**




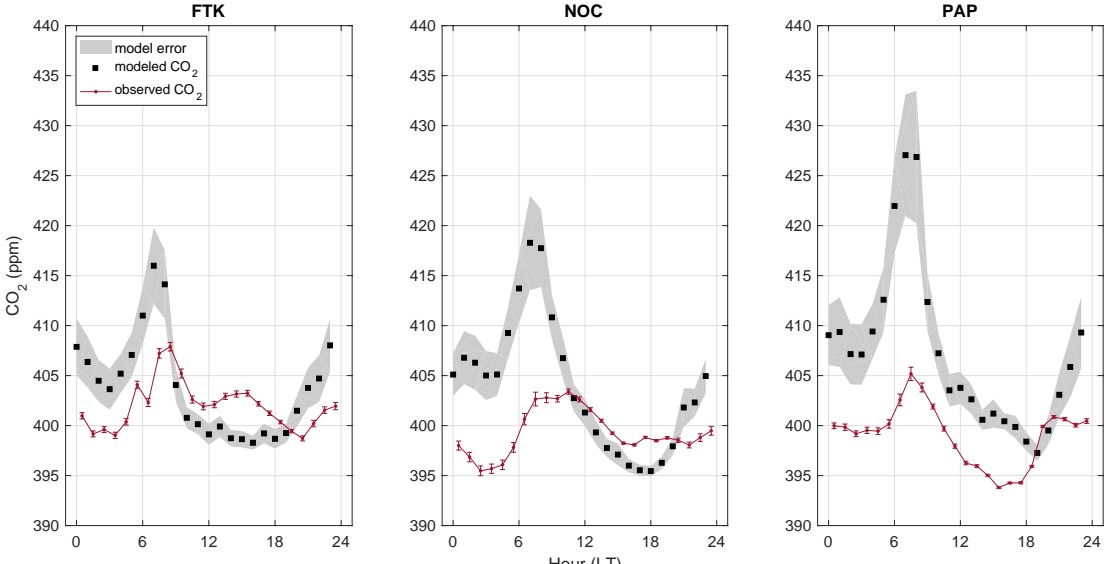

**Figure 11: Diurnal variation in bias-corrected $CO_2$ dry air mole fractions observed and modeled at three representative BEACO₂N sites during September 2013. Error bars and thick shaded curves indicate standard error.**

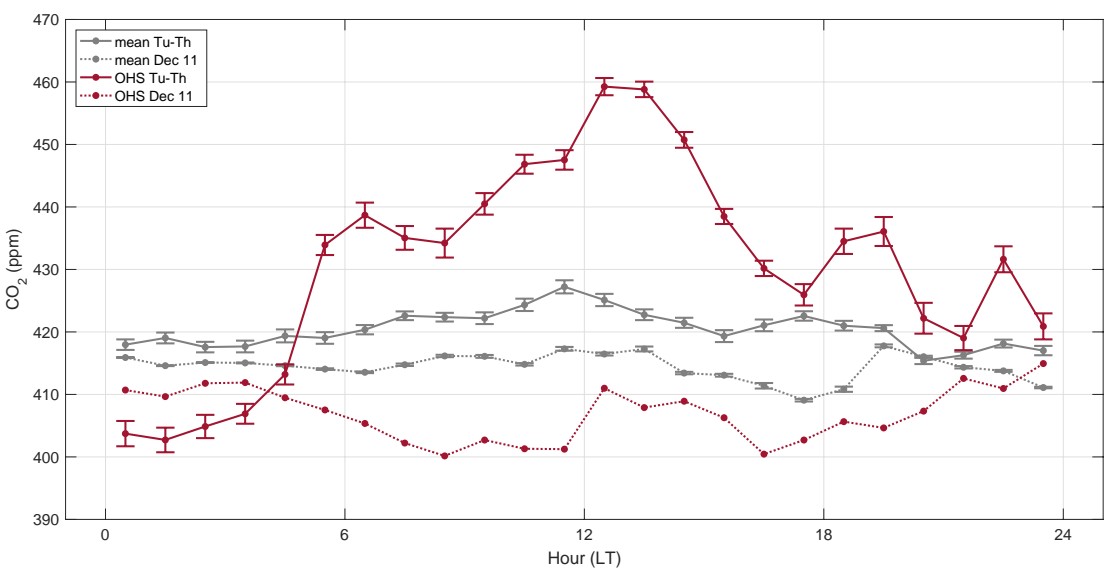

**Figure 12. Comparison of diurnal variation in bias-corrected $CO_2$ dry air mole fractions observed at Oakland High School (OHS in Fig. 1) during a rain-related school closure on 11 December 2014 vs. the mean variation observed on other Tuesdays, Wednesdays, and Thursdays during December 2014 when the school was operating normally. Mean values from five other BEACO₂N sites operational during these time periods are also shown for reference. Error bars indicate standard error.**



| CODE | FULL SITE NAME | LAT | LON | ELEV (m asl) | ELEV (m agl) |
|---|---|---|---|---|---|
| BEL | Burckhalter Elementary School | 37.775 | -122.167 | 97 | 8 |
| BOD | Bishop O'Dowd High School | 37.753 | -122.155 | 82 | 8 |
| CHA | Chabot Space & Science Center | 37.819 | -122.181 | 476 | 11 |
| CPS | College Preparatory School | 37.849 | -122.241 | 102 | 4 |
| EBM | W. Oakland EBMUD Monitoring Stn. | 37.814 | -122.282 | 6 | 2 |
| ELC | El Cerrito High School | 37.907 | -122.294 | 49 | 13 |
| EXB | Exploratorium (Bay) | 37.802 | -122.397 | 13 | 9 |
| EXE | Exploratorium (Embarcadero) | 37.801 | -122.399 | 13 | 5 |
| FTK | Fred T. Korematsu Discovery Acad. | 37.737 | -122.173 | 16 | 6 |
| HRS | Head Royce School | 37.809 | -122.204 | 114 | 5 |
| ICS | International Community School | 37.779 | -122.231 | 19 | 6 |
| KAI | Kaiser Center | 37.809 | -122.264 | 115 | 111 |
| LAU | Laurel Elementary School | 37.792 | -122.196 | 74 | 6 |
| LBL | Lawrence Berkeley Nat'l Lab, Bldg. 70 | 37.876 | -122.252 | 246 | 11 |
| LCC | Lighthouse Community Charter School | 37.736 | -122.196 | 9 | 5 |
| MAR | Berkeley Marina | 37.863 | -122.314 | 6 | 2 |
| MON | Montclair Elementary School | 37.830 | -122.211 | 193 | 4 |
| NOC | N. Oakland Community Charter School | 37.833 | -122.277 | 24 | 6 |
| OHS | Oakland High School | 37.805 | -122.236 | 49 | 7 |
| PAP | PLACE at Prescott Elementary | 37.809 | -122.298 | 12 | 6 |
| PDS | Park Day School | 37.832 | -122.257 | 39 | 7 |
| PHS | Piedmont Middle & High School | 37.824 | -122.233 | 86 | 10 |
| POR | Port of Oakland Headquarters | 37.796 | -122.279 | 35 | 32 |
| ROS | Rosa Parks Elementary School | 37.865 | -122.295 | 22 | 10 |
| SET | Stone Edge Farms (near turbine) | 38.289 | -122.503 | 54 | 2 |
| SEV | Stone Edge Farms (in vineyard) | 38.291 | -122.506 | 61 | 3 |
| SHS | Skyline High School | 37.798 | -122.161 | 359 | 3 |
| STL | St. Elizabeth High School | 37.779 | -122.222 | 28 | 11 |

Table 1: List of site names, abbreviated codes, geo-coordinates, and elevations.





| SITE CODE | UPTIME (%) | MEAN (ppm) | STD (ppm) | MAX (ppm) | MIN (ppm) |
|---|---|---|---|---|---|
| CPS | 94.6 | 416 | 21.6 | 589 | 385 |
|  |  | 423 | 24.3 | 730 | 384 |
| ELC | 90.1 | 411 | 18.5 | 581 | 387 |
|  |  | 415 | 21.3 | 567 | 388 |
| FTK | 91.2 | 415 | 17.7 | 609 | 387 |
|  |  | 418 | 26.4 | 567 | 383 |
| HRS | 69.1 | 410 | 14.7 | 506 | 384 |
|  |  | 428 | 18.4 | 514 | 398 |
| LAU | 91.2 | 429 | 22.4 | 687 | 392 |
|  |  | 421 | 26.4 | 603 | 381 |
| KAI | 83.1 | 442 | 21.8 | 820 | 396 |
|  |  | 418 | 24.7 | 604 | 382 |
| NOC | 87.3 | 411 | 18.4 | 560 | 387 |
|  |  | 428 | 50.5 | 724 | 384 |
| PAP | 55.5 | 403 | 9.57 | 500 | 387 |
|  |  | 411 | 19.1 | 548 | 388 |
| STL | 59.1 | 417 | 17.5 | 586 | 389 |
|  |  | 421 | 36.9 | 616 | 383 |

**Table 2: Descriptive statistics for the bias-corrected $CO_2$ dry air mole fraction measured at nine representative sites during 2013. Upper row for each site gives the daytime (1100–1800 LT) statistics; lower row gives the nighttime (2200–0400 LT). The ELC node is used as the reference site in Sect. 3.4 and so is presented here without bias correction.**





| SITE CODE | $[CO_2]_{\text{temporal drift}}$ day$^{-1}$ (ppm day$^{-1}$) | $[CO_2]_{\text{atemporal bias}}$ (ppm) |
|:---:|:---:|:---:|
| BEL | 0.027 | -2.7 |
| CPS | -0.014 | 3.1 |
| FTK | 0.022 | 6.1 |
| HRS | -0.12 | 1.2 |
| LAU | 0.097 | -27 |
| KAI | 0.042; -0.079 | -23; 5.8 |
| NOC | -0.11; 0.030 | 22; -2.7 |
| PAP | -0.092 | 8.7 |
| STL | -0.034 | 9.4 |

**Table 3: Results from bias-correction analysis at sites for which at least 3 months of observations are available for comparison with the ELC BEACO$_2$N node.**