# Peer review of "The BErkeley Atmospheric CO2 Observation Network: initial evaluation"

_Atmospheric Chemistry and Physics, 2016_

## Referee Comment (RC1) · Anonymous Referee #1 · 19 Jul 2016

This manuscript presents an overview of a city-scale CO2 monitoring network, based on low-cost instruments. The manuscript is concise and well written and presents an interesting experiment. The instrumentation described appears to be well designed and mostly well tested. My primary concerns center on the setup of the network as a whole, the uncertainty quantification and the interpretation of what is or is not possible with a network of this nature.

General comments:

1. In an urban network such as this, the details of how the instruments are situated is likely to be critical. However, only a very cursory explanation is given in the text (P5, L28: 2 to 111 m above ground level. . .). I find it a little concerning that the instruments appear to be situated either very close to the ground (2m), or on rooftops. In urban

areas, flows will be significantly influenced by near-by obstacles within the "roughness sublayer". It is typically assumed that this layer extends at least 2 building heights into the atmosphere (Roth et al., 2000). This is important for emissions verification, because: a) in order to calculate fluxes from concentration measurements, we need to be able to accurately simulate flows from source to receptor; b) measurements within the roughness sublayer, or worse, within the urban canopy layer, will be representative of only a very small area around them, rather than the wider region. If the BEACON instruments are all within the roughness sublayer, I suspect that the network will not be able to meet its aim of monitoring changes in city-scale fluxes, because changes will be representative of only very localized areas, and the modelling requirements of simulating complex flows around buildings, etc., will be too demanding (see comment regarding Figure 12). The authors need to provide much more detail on how they plan to deal with these issues, and whether the instruments have already been situated to account for these factors.

2. I think that the discussion of uncertainties in the instrumentation could be expanded upon further. In particular, I would like to see further characterization of instrumental drift. More details are given below.

3. A network such as this is an exciting and important development. However, I think the authors should be a little more self-critical about the potential limitations. In particular, claims such as a 2% potential accuracy on emissions estimates seem overly optimistic to me for reasons given below.

Specific comments:

P1 L30: I think it's a bit of an exaggeration to say that national monitoring networks "give no information" on urban emissions. A network of instruments in rural areas can still see integrated signals from nearby cities.

P3 L17: With the assertion that uncertainties scale with sqrt(N), the authors are making the assumption that each sensor is an independent estimate of the city-wide concen-

tration. Whilst I agree with the sentiment that an increased number of (well sited and modelled) instruments would lead to a decrease in uncertainty, I suspect that the uncertainty reduction on the urban scale will be nowhere near sqrt(N), which must be considered a theoretical limit. In reality, each instrument will see a footprint around it on the order of a few km, superimposed on some signal from the wider region. Even in a somewhat more "box-model"-like limit, the sqrt(N) argument would assume that all sensors "see" the entire integrated signal of the city, whereas in reality, they would only see everything upstream. Therefore, at any one time, only some subset of the network will be seeing anything close to the "whole" city.

Section 1.4 and Section 3.4: I'm not sure if the term "bias" is the most appropriate here. It appears from Rigby et al. (2008) and section 3.4 that in addition to potential offsets (which I would class as a bias), these instruments can also drift on a range of timescales. Perhaps "systematic uncertainties" would be more appropriate? Furthermore, I think that the assertion that the instruments can be considered "unbiased" if any systematic uncertainty is smaller than the precision is a little difficult. In practice, any systematic offsets could be much more critical than the repeatability. Even relatively small values could have a major impact on an inversion. I think that the paper would benefit from a more nuanced approach in which the uncertainties are more fully characterized. In particular, I think a discussion of the potential "uncorrected" instrumental drift should be shown (i.e. the authors present a method for correcting ∼weekly drifts. However, to compare the data to a model, one would still need to know what the magnitude of potential sub-weekly drift is likely to be).

P5 L14: Following from the discussion above, I think that this comparison would benefit from a plot of the time-varying difference between the two instruments to assess the magnitude of the drift that one would expect in the field.

P6 L16: The running costs for a CRDS seem very high here. Furthermore, the sentence sounds like pumps, data loggers, etc are "annual" running costs, which seems erroneous to me.

P7 L4: How has the influence of wind speed and boundary layer height been isolated from other factors? Surely boundary layer height will be strongly correlated with e.g. an emissions diurnal cycle?

P7 L31: As I understand it, the correction for weekly drift makes the implicit assumption that all sites see the same minimum $CO_2$ concentration as the reference sites. Can the authors comment on how robust this assumption is likely to be? I'm particularly concerned that, with a vertical difference of 500masl between the sensors, this procedure could add biases into the network due to persistent vertical gradients within the network in a particular week.

P1, first paragraph: The discussion of uncertainty reduction largely focuses on another paper under review (Turner et al., 2016). However, I suspect that the estimate that the "accuracy" of Oakland emissions could be reduced to less than 2% (or even 18%) is wildly optimistic for the following reasons: a) Synthetic data studies of this nature make heavy assumptions of Gaussian PDFs and unbiased statistics; b) systematic model errors are largely ignored. In reality, my suspicion is that models will have a very tough job of accurately simulating flows at these scales. I do not agree with the assertion that "These combined error budgets are typically dominated by transport (model) error, which potentially explains why models based on BEACO2N-like networks perform comparably to or better than those based on sparser networks of higher quality sensors, for which instrument error may be reduced but accurately representing transport between observation sites is of greater importance." I suspect that, given the resolution of the flows involved, it may be even more difficult for a model to accurately simulate concentrations for dense urban monitoring network at present (such difficulties would be impossible to discern in a synthetic data experiment). Furthermore, uncertainties in inversions such as this are likely to be very non-Gaussian, and I suspect that the uncertainty budget is likely to be dominated by systematic factors in both the observations and the model.

Figure 11: This figure appears to show model simulations at three sites. However, no

details of the model are given in the text. Either the model setup should be explained, or the model runs should be removed from the figure.

Figure 12: To me, this figure of a site in a school suggests that representation issues in the current network could be severe. The authors show that concentrations were substantially lower on a day when the school was closed. The magnitude of the signal ($\sim$50ppm) shows that this sensor must be completely dominated by the school. Therefore, can we not conclude that the sensor sees little of the wider city, and any long-term changes in concentration at this location will be indicative primarily of change in the school's emissions? Certainly, separating a city-wide 65ppb decrease from this signal (1000x smaller) would seem highly challenging.

Additional References

Roth, M. (2000) Review of atmospheric turbulence over cities, Quarterly Journal of the Royal Meteorological Society, 126, 941-990.

---

## Referee Comment (RC2) · Anonymous Referee #2 · 2 Aug 2016

The manuscript describes a novel CO2 monitoring network consisting of low-cost sensors. First results from a measurement period are presented. The manuscript is well written, with mostly well-prepared figures and a clear structure. I recommend publication after the following comments have been addressed.

General comments:

1. The procedure used for bias correction is somewhat unclear the way it is described. What I understand: a CO2 background determined as the weekly minimum at a reference site is subtracted from the CO2 time series at all sites, then each timeseries is deseasonalized, and the weekly minimum of the resulting timeseries is fitted as a piecewise linear function of time to derive the time-dependent part of bias (B_temporal). After removing this time-dependent part of the bias, the mean of the weekly minima at

each site are taken as the constant bias term and subtracted from the data. Overall it looks like a high-pass filtering of the data (after de-seasonalizing), as slowly varying or constant contributions are subtracted. The question is if any contribution of constant gradients between the different stations (as expected given the differences in near-field emissions) are left after this procedure, or if the assumption really is that each site potentially "sees" background air once per week.

2. Error propagation should be included, propagating errors after pressure correction, temperature correction, water vapour correction, and bias removal (time-varying and constant). In that context it is worth mentioning that the bias error is dominant, not the precision error, when aggregating to yearly signals.

I second the referee #1 comment on the model representation error, which is really crucial as only with a transport model the observations can be quantitatively linked to the fluxes that are of interest.

Specific comments:

P1 L23: The reference "A.B. 32, 2006" should probably read "Brown et al., 2006"

P2 L24: Fig. 3 shows gradients in CO2 fluxes, not in CO2

P3 L10-25: "... sensitivity to changes <10 ppm per year are required" this is quite large compared to the 65 ppb/year. On which metric or specification is the focus (mentioned in line 24)?

P7 L15: The precision of 1-minute averages of the Picarro CRDS systems should be lower than 0.1 ppm, as for a single five second measurement is specified by the manufacturer to be better than 70 ppb (25 ppb for 5 min. averages).

P7 L17: Were the different instrument's time response taken into account in the comparison of the CarboCap and the CRDS? As the CarboCap has a diffusion driven exchange of the sample gas, the instrument response function should be quite different from the more or less instantaneous measurement characteristics of the G2301.

Taking this into account would potentially improve the precision estimate based on the comparison.

Fig. 9 should be modified, as it is impossible to discern the different time series. May be a series of time series plots with 3-5 sites per plot, all sharing the same time axis, but with different vertical ($CO_2$) axis.

P9 L25-28: the fact that the seasonal cycles agree in summer and not in winter seems mostly related to the choice of July as a reference

Caption figure 11: is the "standard error" the error of the mean, or the standard deviation? This should be made clear.

P10 L9: what was used as lateral boundary condition for the regional WRF-STILT model? This is not specified in the Turner et al. (2016) paper focused on network design.
* * *

---

## Author Comment (AC1) · 9 Sep 2016

We thank the referee for their comments and time, which have improved our manuscript in many ways, as detailed below:

*This manuscript presents an overview of a city-scale CO2 monitoring network, based on low-cost instruments. The manuscript is concise and well written and presents an interesting experiment. The instrumentation described appears to be well designed and mostly well tested. My primary concerns center on the setup of the network as a whole, the uncertainty quantification and the interpretation of what is or is not possible with a network of this nature.*

*1. In an urban network such as this, the details of how the instruments are situated is likely to be critical. However, only a very cursory explanation is given in the text (P5, L28: 2 to 111 m above ground level. . .). I find it a little concerning that the instruments appear to be situated either very close to the ground (2m), or on rooftops. In urban areas, flows will be significantly influenced by near-by obstacles within the "roughness sublayer". It is typically assumed that this layer extends at least 2 building heights into the atmosphere (Roth et al., 2000). This is important for emissions verification, because: a) in order to calculate fluxes from concentration measurements, we need to be able to accurately simulate flows from source to receptor; b) measurements within the roughness sublayer, or worse, within the urban canopy layer, will be representative of only a very small area around them, rather than the wider region. If the BEACON instruments are all within the roughness sublayer, I suspect that the network will not be able to meet its aim of monitoring changes in city-scale fluxes, because changes will be representative of only very localized areas, and the modelling requirements of simulating complex flows around buildings, etc., will be too demanding (see comment regarding Figure 12). The authors need to provide much more detail on how they plan to deal with these issues, and whether the instruments have already been situated to account for these factors.*

We understand the concern regarding the representativeness of sensors located within the roughness sublayer. However, a strong sensitivity to local phenomena does not necessarily preclude a given sensor's ability to provide information about the larger domain. As this network is the first of its kind with such a high quantity and density of sensors, more investigation is certainly required to definitively describe the relationship between local and citywide sensitivity, a full exploration of which is beyond the scope of this paper. In the meantime, we have added and revised language in several places throughout the paper (see below) to clarify the degree of uncertainty surrounding the potential capabilities and applications of these sensors, and look forward to providing more quantitative evidence in future publications.

"This largely opportunistic siting approach avoids the logistical and financial obstacles associated with tall tower sampling mechanisms, although it does present additional challenges for the quantification of network-wide phenomena in that no low-lying instrument can singlehandedly provide sensitivity to the entire domain. Installing sensors near the surface and/or built environment does ensure heightened sensitivity to individual, ground-level emissions phenomena, but it is currently unknown whether a well-reasoned combination of these locally sensitive signals from a high volume of sensors could nonetheless yield reliable information about the integrated region. A full exploration of this possibility is beyond the scope of this study; the following analyses focus instead on establishing BEACO$_2$N as a viable platform for investigating such hypotheses."

"Although BEACO₂N demonstrates sensitivity to both highly local fluctuations as well as slowly-varying hemispheric cycles, how best to bootstrap the network's measurements into the analysis of intermediary mesoscale phenomena remains to be determined. Future work will focus on constructing inferred emissions patterns and trends at this scale from the body of observations. In an initial effort in this regard…"

*2. I think that the discussion of uncertainties in the instrumentation could be expanded upon further. In particular, I would like to see further characterization of instrumental drift. More details are given below*

We have added a more thorough characterization of the uncertainties associated with our instrumentation throughout the text; see our responses to specific comments below for details.

*3. A network such as this is an exciting and important development. However, I think the authors should be a little more self-critical about the potential limitations. In particular, claims such as a 2% potential accuracy on emissions estimates seem overly optimistic to me for reasons given below.*

We have revised language throughout the text to clarify the limitations of our original claims; see our response to general comment #1 above and to the specific comment pertaining to the 2% emissions accuracy claim in particular.

*P1 L30: I think it's a bit of an exaggeration to say that national monitoring networks "give no information" on urban emissions. A network of instruments in rural areas can still see integrated signals from nearby cities.*

We have revised the original language as follows to provide a more nuanced description of the capabilities of national monitoring networks:

"Traditional strategies for assessing greenhouse gas emissions are limited to a small handful of monitoring instruments scattered sparsely across remote areas, mostly in developed nations (e.g., Worthy et al., 2003; Thompson et al., 2009; Andrews et al., 2014). These stations are capable of measuring regional average and some integrated urban concentrations with extreme accuracy and precision, but are purposefully distanced from and experience reduced sensitivity to urban signals, thus giving little to no spatially resolved information on emissions in the precise areas that the majority of greenhouse gas rules aim to regulate."

*P3 L17: With the assertion that uncertainties scale with sqrt(N), the authors are making the assumption that each sensor is an independent estimate of the city-wide concentration. Whilst I agree with the sentiment that an increased number of (well sited and modelled) instruments would lead to a decrease in uncertainty, I suspect that the uncertainty reduction on the urban scale will be nowhere near sqrt(N), which must be considered a theoretical limit. In reality, each instrument will see a footprint around it on the order of a few km, superimposed on some signal from the wider region. Even in a somewhat more "box-model"-like limit, the sqrt(N) argument would assume that all sensors "see" the entire integrated signal of the city, whereas in reality, they would*

*only see everything upstream. Therefore, at any one time, only some subset of the network will be seeing anything close to the "whole" city.*

We agree that a sqrt(N) improvement in uncertainty is a perhaps unreasonable theoretical limit, and have revised the text as follows:

"If the goal is verification of regional inter-annual emissions targets, we would therefore require *N* instruments of sufficient individual sensitivity and spatial representativeness such that their combined signals allow us to detect annual changes of ~65 ppb year$^{-1}$ with confidence."

*Section 1.4 and Section 3.4: I'm not sure if the term "bias" is the most appropriate here. It appears from Rigby et al. (2008) and section 3.4 that in addition to potential offsets (which I would class as a bias), these instruments can also drift on a range of timescales. Perhaps "systematic uncertainties" would be more appropriate? Furthermore, I think that the assertion that the instruments can be considered "unbiased" if any systematic uncertainty is smaller than the precision is a little difficult. In practice, any systematic offsets could be much more critical than the repeatability. Even relatively small values could have a major impact on an inversion. I think that the paper would benefit from a more nuanced approach in which the uncertainties are more fully characterized. In particular, I think a discussion of the potential "uncorrected" instrumental drift should be shown (i.e. the authors present a method for correcting ~weekly drifts. However, to compare the data to a model, one would still need to know what the magnitude of potential sub-weekly drift is likely to be).*

We have revised language throughout the original text to characterize temporal drift as a "systematic uncertainty" rather than a "bias" and avoid spurious comparisons between the magnitude of these systematic uncertainties and that of the precision. We have also added more detailed accounting of uncertainties throughout the text, including an exploration of sub-weekly drift, as follows:

"…the standard deviation of their differences is tightened from ±1.5 ppm to ±1.4 ppm. This still exceeds the ±1.0 ppm precision one would expect under average conditions given the form of Eq. (1) and (2) and the manufacturer's specifications for the meteorological sensors (see Sect. 3.5), the CarboCap, and the Picarro (Sect. 3.3), suggesting that the combined effect of the lingering temperature and water biases with any unknown factors is ±0.4 ppm."

"Also presented in Fig. 5 is a time series of the running 1 hour means of the differences between the minute-averaged CarboCap and Picarro observations, demonstrating a short-term drift incurred on approximately hourly timescales found to range between 0.01 and 2.9 ppm during any given 6 hour period of the co-location. The upper bound exceeds the ±1 ppm manufacturer–specified 6 hour short-term stability as well as the 1.5 ppm maximum short-term drift observed by Rigby et al. (2008), but in many cases longer averaging times can be used to reduce the influence of short-term drift to below 1 ppm. Some modelling studies, for example, utilize time steps of 6 hours or more (e.g. Bréon et al., 2015; Wu et al., 2016), and average diurnal cycles can often be assessed across several days. Although some applications require finer temporal resolution, these are typically plume-based analyses that rely on rapidly-varying enhancements above recent background concentrations, essentially eliminating concerns about short-term drift."

"Uncertainties in $U_{temporal}$ and $U_{atemporal}$ shown in Table 3 are calculated given ±1.4 ppm random error in the 1 minute averages, ±2.9 ppm short-term drift, and ±2 ppm agreement with the reference site's weekly minima, assumed to add in quadrature. Mapped onto the observations, these uncertainties result in a mean 1 minute error of ±4 ppm. This is the assumed cumulative error used in this study, although longer averaging times could be used to reduce this figure."

Figure 5 and Table 3 have also been updated accordingly.

*P5 L14: Following from the discussion above, I think that this comparison would benefit from a plot of the time-varying difference between the two instruments to assess the magnitude of the drift that one would expect in the field.*

We have added such a subplot to Fig. 5 in the revised manuscript to aid in the more detailed discussion of instrumental uncertainties described above.

*P6 L16: The running costs for a CRDS seem very high here. Furthermore, the sentence sounds like pumps, data loggers, etc are "annual" running costs, which seems erroneous to me.*

The characterization of pumps, data loggers, etc. as "annual" costs is indeed erroneous, and the original language has been revised accordingly:

"For comparison, a single commercial cavity ring-down analyzer is priced around $60,000 USD and the total equipment cost can exceed $85,000 USD after accounting for pumps, data loggers, etc."

*P7 L4: How has the influence of wind speed and boundary layer height been isolated from other factors? Surely boundary layer height will be strongly correlated with e.g. an emissions diurnal cycle?*

We agree that the existing analysis does not allow us to discriminate between wind speed/boundary layer height fluctuations and other diurnal patterns (e.g. in emissions); and have revised the original language as follows:

"The two sensors nonetheless demonstrate remarkable agreement; while typical diurnal $CO_2$ variations during the same period are on the order of 20–60 ppm, the CarboCaps simultaneously detect $CO_2$ events as small as 8 ppm, providing preliminary evidence of the suitability of these sensors for high-density urban deployment."

*P7 L31: As I understand it, the correction for weekly drift makes the implicit assumption that all sites see the same minimum CO2 concentration as the reference sites. Can the authors comment on how robust this assumption is likely to be? I'm particularly concerned that, with a vertical difference of 500masl between the sensors, this procedure could add biases into the network due to persistent vertical gradients within the network in a particular week.*

We developed this assumption after preliminary analyses revealed that the weekly minima measured at each site roughly tracked the three-dimensional Pacific boundary "curtain" mentioned in the text. If each site (including the reference) samples background air approximately once a week, the sites' weekly minima should agree with one another as well. Such comparison with the boundary curtain is of course complicated by the very drift and biases that the subsequent correction procedure aims to remove, so it is not possible to quantify the influence of vertical gradients independently of systematic uncertainties via this method. We did, however, perform a similar comparison using measurements from the sea level LI-COR LI-820 maintained by the Pacific Marine Environmental Laboratory. As mentioned in the text, this instrument is calibrated against a reference gas prior to every measurement, and so is assumed here to be free of drift and/or bias. Although the LI-820's weekly minima do not agree precisely with the boundary curtain's (residuals ranged from 0–12.7 ppm, with a mean of 1.8 ppm), the deviations from the curtain values were not significantly autocorrelated on timescales greater than one week. From this we conclude that, while the assumption of agreement with a reference site may not be guaranteed for any particular week, the deviations are not in fact persistent across multiple weeks, even for instruments sited well within the roughness sublayer. Because we require at least three months of comparison with the reference for drift correction, the influence of anomalous weeks is minimized, and we are confident that, on these timescales, network-wide weekly minima agree to within approximately ±2 ppm. We have added an explanation of the preceding comments to the text as follows:

"BEACO$_2$N's unique location near the Pacific coast results in a relatively consistent wind direction from largely unpolluted over-ocean origins, such that the weekly minima can be assumed to reflect both the seasonal and synoptic variations in network-wide baseline CO$_2$ concentrations while avoiding the influence of shorter term variability in local sources and sinks. This assumption is supported by preliminary analyses comparing observations from a LI-COR LI-820 non-dispersive infrared CO$_2$ gas analyzer with a smoothed, three-dimensional "curtain" of surface CO$_2$ Pacific boundary conditions produced by NOAA's Global Greenhouse Gas Reference Network (Jeong et al., 2013). The LI-COR, positioned at sea level between the EXB and EXE nodes (see Fig. 1), is maintained by NOAA's Pacific Marine Environmental Laboratory and calibrated against compressed gas (400–500 ppm CO$_2$) prior to every hourly measurement and is assumed to have negligible bias. Despite a proximity to local surface-level emissions and complex boundary layer dynamics, the LI-COR's weekly minima are found to generally follow variations in the Pacific curtain, with an average residual of ~2 ppm."

*P1, first paragraph: The discussion of uncertainty reduction largely focuses on another paper under review (Turner et al., 2016). However, I suspect that the estimate that the "accuracy" of Oakland emissions could be reduced to less than 2% (or even 18%) is wildly optimistic for the following reasons: a) Synthetic data studies of this nature make heavy assumptions of Gaussian PDFs and unbiased statistics; b) systematic model errors are largely ignored. In reality, my suspicion is that models will have a very tough job of accurately simulating flows at these scales. I do not agree with the assertion that "These combined error budgets are typically dominated by transport (model) error, which potentially explains why models based on BEACO2N-like networks perform comparably to or better than those based on sparser networks of higher quality sensors, for which instrument error may be reduced but accurately representing transport between observation sites is of greater importance." I suspect that, given the resolution of the flows*

*involved, it may be even more difficult for a model to accurately simulate concentrations for dense urban monitoring network at present (such difficulties would be impossible to discern in a synthetic data experiment). Furthermore, uncertainties in inversions such as this are likely to be very non-Gaussian, and I suspect that the uncertainty budget is likely to be dominated by systematic factors in both the observations and the model.*

The referee raises multiple important points:

Turner et al. (2016) do assume Gaussian PDFs and the error statistics may not in fact be Gaussian. However, the true form of the distribution is typically unknown. Additionally, the assumption of Gaussian prior and likelihood distributions has the benefit of the posterior and prior being conjugate distributions. This means that we have a closed-form expression for the posterior distribution and we can decompose the prior covariance matrix with a Kronecker product (e.g., Yadav & Michalak, 2013; Sect. S2 in Turner et al., 2016). All of this allows us to construct high-dimensional state vectors with fully populated covariance matrices that will give us a better representation of the true statistics than assuming, say, uncorrelated errors. In the absence of knowledge of the true distribution, Gaussian distribution seem to be a fair assumption.

With regards to the effect of systematic errors, these are not ignored, but discussed in Sections 6.1 and S6.2 of Turner et al. (2016).

In terms of the error budget in general and transport error in particular, we agree that the flows will be more difficult to correctly simulate at high resolution in an urban region and will induce a large uncertainty. This term is probably the largest uncertainty, as stated in the text. However, the referee speculates that it would be more difficult to simulate concentrations for a dense network (presumably, compared to a sparse network). This is a point that is discussed in Section 6.3 in Turner et al. (2016):

"In this work we have treated transport error and the number of measurement sites as independent. However, in practice, there would be a relationship between the transport error and measurement network density. This can be understood with a thought experiment using two different observing systems to estimate emissions: a sparse network with a single site and an infinitely dense network (sites at each grid cell in our domain). Estimating emissions with the sparse network would require us to simulate the atmospheric transport with high fidelity if we are to reliably say anything about emissions upwind of our site. This is especially true for point sources. Any errors in the simulated atmospheric transport would adversely impact the estimated emissions, whereas the infinitely dense network could potentially neglect atmospheric transport and use data from only the local grid cell to estimate emissions. This is because the differential signal at each site would be largely governed by the local emissions."

*Figure 11: This figure appears to show model simulations at three sites. However, no details of the model are given in the text. Either the model setup should be explained, or the model runs should be removed from the figure.*

Additional details regarding the model setup have been added to the text as follows:

"We simulate hourly $CO_2$ concentrations ($\widehat{\boldsymbol{y}}$) at each site in the network using the Stochastic Time-Inverted Lagrangian Transport model (STILT; Lin et al., 2003) coupled to the Weather Research and Forecasting model (WRF; Skamarock et al., 2008). The coupled model is known as "WRF-STILT" (Nehrkorn et al. 2010) and the setup used here follows that of Turner et al. (2016; see their Sect. S1 for details of the WRF setup). WRF-STILT advects an ensemble of 500 particles 3 days backwards in time, each with a small random perturbation, from the spatio-temporal locations of the BEACO$_2$N observations using the meteorological fields from WRF. The trajectories of these 500 particles are then used to construct "footprints" for each observation that represent the sensitivity of the observation to a perturbation in emissions from a given location. The footprints can be represented in matrix form ($\boldsymbol{H}$) and multiplied by a set of gridded emissions ($\boldsymbol{x}$, from the high-resolution bottom-up $CO_2$ inventory in Turner et al. 2016) to compute the $CO_2$ enhancement at each site due to local emissions:

$$\Delta\boldsymbol{y} = \boldsymbol{Hx} \qquad\qquad\qquad (5)$$

We then add this local enhancement to a background concentration ($\boldsymbol{y_B}$, from the aforementioned Pacific boundary curtain) to obtain a model estimate of the BEACO$_2$N observations shown as black squares in Fig. 11:

$$\widehat{\boldsymbol{y}} = \Delta\boldsymbol{y} + \boldsymbol{y_B} = \boldsymbol{Hx} + \boldsymbol{y_B} \qquad\qquad\qquad (6)"$$

*Figure 12: To me, this figure of a site in a school suggests that representation issues in the current network could be severe. The authors show that concentrations were substantially lower on a day when the school was closed. The magnitude of the signal (~50ppm) shows that this sensor must be completely dominated by the school. Therefore, can we not conclude that the sensor sees little of the wider city, and any long-term changes in concentration at this location will be indicative primarily of change in the school's emissions? Certainly, separating a city-wide 65ppb decrease from this signal (1000x smaller) would seem highly challenging.*

This figure does demonstrate one sensor's strong sensitivity to its local environment, however, as mentioned earlier in our response, there is not yet any evidence to suggest that a sensor's local sensitivity is necessarily mutually exclusive with its utility in assessing domain-wide phenomena, although further investigation is clearly needed in this area. We hope that our aforementioned revisions describing the uncertainties surrounding this issue of sensitivity will help to assuage concerns about this figure in particular.

*References:*

Turner, A. J., Shusterman, A. A., McDonald, B. C., Teige, V., Harley, R. A., and Cohen. R. C.: Network design for quantifying urban CO2 emissions: Assessing trade-offs between precision and network density, Atmos. Chem. Phys. Discuss., in review, doi:10.5194/acp-2016-355, 2016.

Yadav, V. and Michalak, A. M.: Improving computational efficiency in large linear inverse problems: an example from carbon dioxide flux estimation, Geosci. Model Dev., 6, 583–590, doi:10.5194/gmd-6-583-2013, 2013.

---

## Author Comment (AC2) · 9 Sep 2016

We appreciate the referee's time and feedback, which have resulted in significant improvements to our manuscript, as detailed below:

*The manuscript describes a novel CO2 monitoring network consisting of low-cost sensors. First results from a measurement period are presented. The manuscript is well written, with mostly well-prepared figures and a clear structure. I recommend publication after the following comments have been addressed.*

*1. The procedure used for bias correction is somewhat unclear the way it is described. What I understand: a CO2 background determined as the weekly minimum at a reference site is subtracted from the CO2 time series at all sites, then each timeseries is deseasonalized, and the weekly minimum of the resulting timeseries is fitted as a piecewise linear function of time to derive the time-dependent part of bias (B_temporal). After removing this time-dependent part of the bias, the mean of the weekly minima at each site are taken as the constant bias term and subtracted from the data. Overall it looks like a high-pass filtering of the data (after de-seasonalizing), as slowly varying or constant contributions are subtracted. The question is if any contribution of constant gradients between the different stations (as expected given the differences in near- field emissions) are left after this procedure, or if the assumption really is that each site potentially "sees" background air once per week.*

Because the weekly minima measured at the reference site will reflect the effective seasonal variation present in the BEACO$_2$N domain, the initial "background subtraction" step and the de-seasonalization are one and the same. To clarify this point, we have revised the text as follows:

"Once the $[CO_2]_{background}$ term is removed, effectively de-seasonalizing the observations, we re-calculate the weekly minima of this new data record and fit the result as a (piecewise, if necessary) linear function of time…"

Furthermore, we have reason to believe that each site samples background air approximately once per week. As described in our response to Referee #1, preliminary analyses revealed that the weekly minima measured at each site roughly track the three-dimensional Pacific boundary "curtain" mentioned in the text. Such comparison with the boundary curtain is of course complicated by the very drift and biases that the subsequent correction procedure aims to remove, so it is not possible to quantify the influence of inter-site gradients independently of systematic uncertainties via this method. We did, however, perform a similar comparison using measurements from the sea level LI-COR LI-820 maintained by the Pacific Marine Environmental Laboratory. As mentioned in the text, this instrument is calibrated against a reference gas prior to every measurement, and so is assumed here to be free of drift and/or bias. Although the LI-820's weekly minima do not agree precisely with the boundary curtain's (residuals ranged from 0–12.7 ppm, with a mean of 1.8 ppm), the deviations from the curtain values were not significantly autocorrelated on timescales greater than one week. From this we conclude that, while the assumption of agreement with a reference site may not be guaranteed for any particular week, the deviations are not in fact persistent across multiple weeks, even for instruments sited well within the roughness sublayer. Because we require at least three months of comparison with the reference for drift correction, the influence of anomalous weeks is minimized, and we are confident that, on these timescales, the network-wide weekly minima agree to within approximately ±2 ppm. This

explanation has been added to the text (see below), and the uncertainty associated with this assertion has been factored into a more detailed error propagation (described later in this response).

"BEACO₂N's unique location near the Pacific coast results in a relatively consistent wind direction from largely unpolluted over-ocean origins, such that the weekly minima can be assumed to reflect both the seasonal and synoptic variations in network-wide baseline $CO_2$ concentrations while avoiding the influence of shorter term variability in local sources and sinks. This assumption is supported by preliminary analyses comparing observations from a LI-COR LI-820 non-dispersive infrared $CO_2$ gas analyzer with a smoothed, three-dimensional "curtain" of surface $CO_2$ Pacific boundary conditions produced by NOAA's Global Greenhouse Gas Reference Network (Jeong et al., 2013). The LI-COR, positioned at sea level between the EXB and EXE nodes (see Fig. 1), is maintained by NOAA's Pacific Marine Environmental Laboratory and calibrated against compressed gas (400–500 ppm $CO_2$) prior to every hourly measurement and is assumed to have negligible bias. Despite a proximity to local surface-level emissions and complex boundary layer dynamics, the LI-COR's weekly minima are found to generally follow variations in the Pacific curtain, with an average residual of ~2 ppm."

*2. Error propagation should be included, propagating errors after pressure correction, temperature correction, water vapour correction, and bias removal (time-varying and constant). In that context it is worth mentioning that the bias error is dominant, not the precision error, when aggregating to yearly signals.*

We have added a more detailed accounting of our error propagation in several places throughout the text, including a consideration of short-term drift, as follows:

"…the standard deviation of their differences is tightened from ±1.5 ppm to ±1.4 ppm. This still exceeds the ±1.0 ppm precision one would expect under average conditions given the form of Eq. (1) and (2) and the manufacturer's specifications for the meteorological sensors (see Sect. 3.5), the CarboCap, and the Picarro (Sect. 3.3), suggesting that the combined effect of the lingering temperature and water biases with any unknown factors is ±0.4 ppm."

"Also presented in Fig. 5 is a time series of the running 1 hour means of the differences between the minute-averaged CarboCap and Picarro observations, demonstrating a short-term drift incurred on approximately hourly timescales found to range between 0.01 and 2.9 ppm during any given 6 hour period of the co-location. The upper bound exceeds the ±1 ppm manufacturer–specified 6 hour short-term stability as well as the 1.5 ppm maximum short-term drift observed by Rigby et al. (2008), but in many cases longer averaging times can be used to reduce the influence of short-term drift to well below 1 ppm. Some modeling studies, for example, utilize time steps of 6 hours or more (e.g. Bréon et al., 2015; Wu et al., 2016), and average diurnal cycles can often be assessed across several days. Although some applications require finer temporal resolution, these are typically plume-based analyses that rely on rapidly-varying enhancements above recent background concentrations, essentially eliminating concerns about short-term drift."

"Uncertainties in $U_{temporal}$ and $U_{atemporal}$ shown in Table 3 are calculated given ±1.4 ppm random error in the 1 minute averages, ±2.9 ppm short-term drift, and ±2 ppm agreement with the reference site's weekly minima, assumed to add in quadrature. Mapped onto the observations, these

uncertainties result in a mean 1 minute error of ±4 ppm. This is the assumed cumulative error used in this study, although longer averaging times could be used to reduce this figure."

Figure 5 and Table 3 have also been updated accordingly.

*3. I second the referee #1 comment on the model representation error, which is really crucial as only with a transport model the observations can be quantitatively linked to the fluxes that are of interest.*

We agree that the surface-level siting of sensors may enhance their sensitivity to local phenomena and limit their single-handed representativeness of the larger domain. However, the individual sensors' local sensitivity does not necessarily prohibit their collective ability to characterize citywide trends and/or events in the context of atmospheric transport models. Turner et al. (2016), for example, demonstrate the utility of synthetic, BEACO$_2$N-like observations in constraining mesoscale fluxes, even in the presence of persistent site-to-site biases. To our knowledge, BEACO$_2$N is the first network with sufficient sensor quantity and density to investigate this issue empirically, a full exploration of which is beyond the scope of this study. To clarify current uncertainties surrounding the representativeness and capabilities of our sensors, we have updated various sections of the text as follows:

"This largely opportunistic siting approach avoids the logistical and financial obstacles associated with tall tower sampling mechanisms, although it does present additional challenges for the quantification of network-wide phenomena in that no low-lying instrument can singlehandedly provide sensitivity to the entire domain. Installing sensors near the surface and/or built environment does ensure heightened sensitivity to individual, ground-level emissions phenomena, but it is currently unknown whether a well-reasoned combination of these locally sensitive signals from a high volume of sensors could nonetheless yield reliable information about the integrated region. A full exploration of this possibility is beyond the scope of this study; the following analyses focus instead on establishing BEACO$_2$N as a viable platform for investigating such hypotheses."

"Although BEACO$_2$N demonstrates sensitivity to both highly local fluctuations as well as slowly-varying hemispheric cycles, how best to bootstrap the network's measurements into the analysis of intermediary mesoscale phenomena remains to be determined. Future work will focus on constructing inferred emissions patterns and trends at this scale from the body of observations. In an initial effort in this regard…"

*P1 L23: The reference "A.B. 32, 2006" should probably read "Brown et al., 2006"*

The Brown et al. publication in the References list refers to California's First Update to the Climate Change Scoping Plan published in 2014, whereas the A.B. 32 citation is intended to refer directly to the California Global Warming Solutions Act passed by the state legislature in 2006 that mandates the creation of such scoping plans. We have revised the format of the latter citation in the References list to clarify this point:

"A.B. 32: California Global Warming Solutions Act, Assemb. Reg. Sess. 2005–2006, (CA 2006)."

*P2 L24: Fig. 3 shows gradients in CO2 fluxes, not in CO2*

The text has been updated to reflect this correction as follows:

"see Fig. 3 for examples of observed intra-city $CO_2$ flux gradients"

*P3 L10-25: ". . . sensitivity to changes <10 ppm per year are required" this is quite large compared to the 65 ppb/year. On which metric or specification is the focus (mentioned in line 24)?*

The focus of the paragraph in question is on the 65 ppb/year metric. The text introducing the <10 ppm/year metric has been revised to clarify as follows:

"The precision requirements at each individual site versus for a network instrument as a whole vary depending on the phenomena of interest. Metropolitan regions produce <10 ppm $CO_2$ enhancements in the boundary layer (Pacala et al., 2010), requiring sensitivity to changes orders of magnitude smaller for the characterization of citywide integrated inter-annual trends, for example."

*P7 L15: The precision of 1-minute averages of the Picarro CRDS systems should be lower than 0.1 ppm, as for a single five second measurement is specified by the manufacturer to be better than 70 ppb (25 ppb for 5 min. averages).*

The ±0.1 ppm precision figure for the Picarro CRDS system was obtained via personal communication with the collaborator who maintains said system, and reflects a combination of the fact that this system is based on an older version of the Picarro G2301 instrument (specified by the manufacturer in 2010 to possess 5 second precision better than 150 ppb) and that it contains a myriad of ancillary custom parts that are not necessarily accounted for in the manufacturer's specifications. This being said, the ±0.1 ppm precision does indeed pertain to a 5 second measurement frequency, not a 1 minute average. We have revised the original text to clarify this fact:

"In this case the 1σ precision of the 1 minute averages is ±1.4 ppm, given by the standard deviation of the differences between the minute-averaged CarboCap and Picarro observations and the Picarro's precision (±0.1 ppm at 5 second measurement frequency)."

*P7 L17: Were the different instrument's time response taken into account in the comparison of the CarboCap and the CRDS? As the CarboCap has a diffusion driven exchange of the sample gas, the instrument response function should be quite different from the more or less instantaneous measurement characteristics of the G2301. Taking this into account would potentially improve the precision estimate based on the comparison.*

The time series are aligned by eye during analysis to account for any lag between the instruments as well as any differences between their timestamp clocks (which was found to be the dominant factor driving the offset between the two datasets). Manual inspection revealed that the various peaks and other structural features in the two time series were of equal duration, giving no

indication of the supposedly slower response time of the CarboCap. This may be partially due to our use of intake/outflow fans, which force air through the sensor faster than it would passively diffuse otherwise. In any case, any small differences in time response that remained after this treatment would be effectively removed by averaging to a one-minute timescale, so this consideration is unlikely to affect our precision estimate.

*Fig. 9 should be modified, as it is impossible to discern the different time series. May be a series of time series plots with 3-5 sites per plot, all sharing the same time axis, but with different vertical (CO2) axis.*

We agree it is difficult to discern specific trends and features from the original version of Fig. 9, but have found the suggested alternative to be similarly inscrutable. Instead, we have chosen to supplement Fig. 9 with additional subplots depicting progressively shorter time periods at a subset of sites. The goal of this figure is simply to qualitatively represent the variety and sheer volume of atmospheric conditions sampled by the $BEACO_2N$ instrument; more quantitative impressions of the dataset are detailed later in the text. Revisions to the original language and caption reflecting the updated version of Fig. 9 can be found below:

"Figure 9 demonstrates the volume and diversity of urban $CO_2$ concentrations sampled, exhibiting extreme short-term variability superimposed on a slower, seasonal fluctuation in the minimum values. For clarity, the bottom panels depicting month- and week-long samples of the overall time series show data from six representative sites. Network-wide, daytime (1100–1800 LT) means between 408 and 442 ppm are observed…"

"Figure 9: Time series of drift- and bias-corrected $CO_2$ dry air mole fractions collected over the course of ~2.5 years at 16 $BEACO_2N$ sites (top), one month at six representative sites (middle), and one week at the same six sites (bottom). The hiatus around Day 600 corresponds to a largescale hardware refurbishment effort that began in mid-2014."

*P9 L25-28: the fact that the seasonal cycles agree in summer and not in winter seems mostly related to the choice of July as a reference*

Here we intend to refer to the magnitude of the seasonal variation—which is independent of the month chosen as reference—rather than the absolute values of the monthly minima. We have revised the language as follows to clarify this point:

"At many sites, the $BEACO_2N$ minima are seen to exhibit a seasonal variation of a magnitude roughly in keeping with that observed in the curtain, while other sites demonstrate a more exaggerated summer-winter contrast, as might be expected within an urban dome."

*Caption figure 11: is the "standard error" the error of the mean, or the standard deviation? This should be made clear.*

The caption has been updated to clarify this point:

"Error bars indicate the standard error of the mean (instrument error is negligible at this timescale); thick shaded curves indicate standard deviation."

*P10 L9: what was used as lateral boundary condition for the regional WRF-STILT model? This is not specified in the Turner et al. (2016) paper focused on network design.*

The lateral boundary conditions for $CO_2$ were provided by NOAA's three-dimensional Pacific boundary "curtain" (Jeong et al., 2013), while the lateral boundary conditions for meteorology were provided by the North American Regional Reanalysis (Mesinger et al., 2006). We have added text specifying the former, as well as text referring the reader to the supplement of Turner et al. (2016) for further details:

"We then add this local enhancement to a background concentration ($\boldsymbol{y_B}$, from the aforementioned Pacific boundary curtain) to obtain a model estimate of the $BEACO_2N$ observations…"

"…the setup used here follows that of Turner et al. (2016; see their Sect. S1 for details of the WRF setup)."

*References:*

Jeong, S., Hsu, Y.-K., Andrews, A. E., Bianco, L., Vaca, P., Wilczak, J. M., and Fischer, M. L.: A multitower measurement network estimate of California's methane emissions, J. Geophys. Res. Atmos., 118, 11339–11351, doi:10.1002/jgrd.50854, 2013.

Mesinger, F., DiMego, G., Kalnay, E., Mitchell, K., Shagran P. C., Ebisuzaki, W., Jović, D., Woollen, J., Rogers, E., Berbery, E. H., Ek, M. H., Fan, Y., Grumbine, R., Higgins, W., Li, H., Lin, Y., Manikin, G., Parrish, D., and Shi, W.: North american regional reanalysis, Bull. Am. Meteorol. Soc., 87, 343–360, doi:10.1175/BAMS-87-3-343, 2006.

Turner, A. J., Shusterman, A. A., McDonald, B. C., Teige, V., Harley, R. A., and Cohen. R. C.: Network design for quantifying urban CO2 emissions: Assessing trade-offs between precision and network density, Atmos. Chem. Phys. Discuss., in review, doi:10.5194/acp-2016-355, 2016.